# Algorithm Discovery With LLMs:
# Evolutionary Search Meets Reinforcement Learning

Anja Surina[1]*, Amin Mansouri[1], Lars Quaedvlieg[1], Amal Seddas[1],
Maryna Viazovska[1], Emmanuel Abbe[1,2], Caglar Gulcehre[1]
[1]EPFL [2]Apple

## Abstract

Discovering efficient algorithms for solving complex problems has been an outstanding challenge in mathematics and computer science, requiring substantial human expertise over the years. Recent advancements in evolutionary search with large language models (LLMs) have shown promise in accelerating the discovery of algorithms across various domains, particularly in mathematics and optimization. However, existing approaches treat the LLM as a static generator, missing the opportunity to update the model with the signal obtained from evolutionary exploration. In this work, we propose to augment LLM-based evolutionary search by continuously refining the search operator – the LLM – through reinforcement learning (RL) fine-tuning. Our method leverages evolutionary search as an exploration strategy to discover improved algorithms, while RL optimizes the LLM policy based on these discoveries. Our experiments on combinatorial optimization tasks demonstrate that integrating RL with evolutionary search accelerates the discovery of superior algorithms, showcasing the potential of RL-enhanced evolutionary strategies for algorithm design.

## 1 Introduction

The ability to solve complex problems efficiently is at the heart of scientific and technological advancement. Whether calculating planetary trajectories, analyzing genomic sequences, ensuring reliable communications; or solving large-scale optimization problems, these challenges require formal and systematic methods to process, analyze, and transform information into decisions by means of *algorithms*. Thus, the history of algorithm design is as ancient as mathematics itself. From early examples like Euclid's algorithm for computing the greatest common divisor and the Sieve of Eratosthenes for identifying prime numbers to modern advancements such as gradient descent (Ruder, 2016) and backpropagation (Rumelhart et al., 1986; Kelley, 1960), the discovery of effective computational methods has consistently shaped the trajectory of science and technology. Despite rapid advancements, the demand for new and efficient algorithms remains strong, as scientific and technological progress continuously presents new challenges. Hence, the impact of well-crafted algorithms make their discovery an everlasting pursuit of significant importance.

Today, this pursuit finds a powerful ally in the unprecedented capabilities of large language models (LLMs). As some of the recent models exhibit reasoning-like behavior (OpenAI, 2024; DeepSeek-AI et al., 2025), a new opportunity arises where LLMs could assist algorithm design, reshaping problem-solving across disciplines. A particularly powerful way to harness such capabilities for algorithm design is through evolutionary search strategies that explore the space of algorithms as executable programs. By combining LLMs with evolutionary search, researchers have achieved remarkable breakthroughs, including discovering novel mathematical constructs that surpass existing knowledge on challenging problems (Romera-Paredes et al., 2024), designing reward functions for training robotic policies (Ma

---

*Correspondence to `anja.surina@epfl.ch`

†We open source our code implementation in `https://claire-labo.github.io/EvoTune/` to facilitate future research in this promising area.

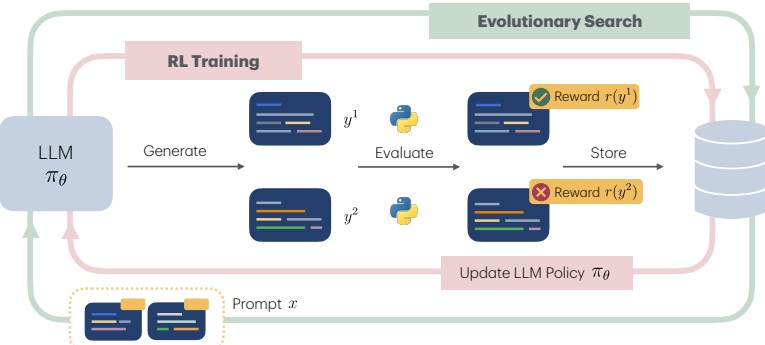

Figure 1: **Method overview:** `EvoTune` iteratively alternates between two phases: (a) *evolutionary search* that iteratively improves solutions by bootstrapping from the best ones discovered so far, and (b) *RL training*, which updates the model parameters based on information gained from the search process. In this loop, evolutionary search is used to explore the space of programs efficiently and collect data, and RL is used to improve the policy based on the data generated with evolutionary search. Python programs generated by an LLM are evaluated on a set of combinatorial optimization problem instances and then stored in a program database for later use in RL training and prompt construction.

et al., 2023), developing preference optimization algorithms (Lu et al., 2024), and outperforming top human teams in combinatorial competitive programming (Veličković et al., 2024).

An influential approach in this line of research is the FunSearch method (Romera-Paredes et al., 2024). Funsearch iteratively proposes new solutions, represented as programs, by combining the most promising programs discovered in earlier iterations. Bootstrapping on previous successes gradually improves the performance of the best program found.

Although FunSearch-like methods achieve impressive results, they regard the LLM as a static generator and do not take advantage of the fact that LLMs are parametric models that can be optimized for specific objectives. Thus, in our work, we propose to augment evolutionary search with LLMs by continuously refining the search operator, the LLM, through RL fine-tuning using feedback from evolutionary exploration. This aligns with the "Bitter Lesson" (Sutton, 2019), which argues that *search* and *learning* are synergistic: Search generates new data, while learning distills patterns from the data to guide future exploration more effectively. Furthermore, most AI systems that have outperformed human performance – from board games (Silver et al., 2016) to real-time strategy games (Vinyals et al., 2019) – have relied on RL in a similar spirit (Sutton & Barto, 2018; Fawzi et al., 2022; Berner et al., 2019), demonstrating the power of this synergy. Using search alone is inefficient and may fail to capture emergent patterns while relying on a fixed dataset for training limits exploration.

Motivated by this gap, we hypothesize that combining reinforcement learning with a FunSearch-like approach can exploit the strengths of both paradigms, enabling more effective algorithm discovery. Similarly to alignment methods (Ouyang et al., 2022; Stiennon et al., 2020), we propose to train the LLM *in-weight* using the evaluation scores of the generated programs as the reward signal. By adding in-weight training, we aim to enable the model to better utilize insights gained from exploration to improve its understanding of the search space and therefore enable better-targeted search at subsequent iterations.

Our contributions are summarized as follows:

- To the best of our knowledge, we are the first to demonstrate the potential of tightly integrating LLM-based evolutionary search with RL in the loop. Our method `EvoTune` uses evolutionary search as an exploration strategy and RL to optimize and improve the policy. For updating the policy, we employ the DPO algorithm (Rafailov et al., 2024); however, in contrast to its standard offline formulation, we leverage it in an off-policy setting with non-fixed inputs.

- By improving the efficiency of the search mechanism, our method accelerates the discovery of superior algorithms. Our experiments spanning three instruction-tuned LLMs demonstrate consistent performance gains over the baseline FunSearch method across a diverse set of benchmarks, including bin packing, the traveling salesman problem, flatpack, Hash Code programming competition problems, and symbolic regression tasks.

- We show the advantage of using a modified version of the standard alignment objective in terms of preserving output diversity, which is critical to the success of evolutionary search strategies.

## 2 Preliminaries

Here, we provide a basic overview of the components needed for the workflow of `EvoTune`. It comprises three components: 1) LLM, 2) evolutionary search, and 3) RL training.

**LLMs** In this work, we use pre-trained LLMs and denote an LLM as $\pi_\theta(\cdot|\cdot)$, which models a conditional distribution $\pi_\theta(y|x)$ autoregressively. $x \in \mathcal{X}$ corresponds to the input prompt and $y \in \mathcal{Y}$ corresponds to the output generated by the model.

**Evolutionary search** Evolutionary search can be defined as an iterative optimization process inspired by biological evolution (Goldberg & Holland, 1988). In the context of LLMs, the goal is to explore the space of LLM-generated programs to maximize a fitness function represented by the reward score $r(x, y)$ (Lehman et al., 2023). The optimization process to find the best output $y^*$ is typically gradient-free, using search heuristics for selection, variation, and diversity maintenance:

$$y^* = \arg\max_y \mathbb{E}_{x \sim \mathcal{D}} \left[ \mathbb{E}_{y \sim \pi_\theta(\cdot|x)}[r(x, y)] \right] . \tag{1}$$

**RL training** RL has proven to be a powerful tool for optimizing policies in complex search spaces, especially when a well-defined reward function is available (Silver et al., 2016; Fawzi et al., 2022; Sutton & Barto, 2018; Vinyals et al., 2019). In `EvoTune`, we integrate RL into the evolutionary search to refine the LLM generation policy over time. Using feedback from the evolutionary search phase, we adapt the parameters of the LLM to generate program candidates that achieve higher performance scores in subsequent search iterations.

To optimize the LLM policy $\pi_\theta$ we can employ an RL objective with regularization to keep the outputs of reference model $\pi_{\mathrm{ref}}(y|x)$ and trained model $\pi_\theta(y|x)$ close to each other:

$$\max_{\pi_\theta} \mathbb{E}_{x \sim \mathcal{D}, y \sim \pi_\theta}[r(x, y)] - \beta \mathbb{D}_f[\pi_{\mathrm{ref}}(\cdot|x)||\pi_\theta(\cdot|x)], \tag{2}$$

where $\mathbb{D}_f$ is an f-divergence typically implemented with reverse KL-divergence, $\beta$ is a hyperparameter controlling the strength of the KL regularization and $\pi_{\mathrm{ref}}$ is the reference policy, in our case the initial policy $\pi_\theta^0$, corresponding to the base LLM. It is possible to maximize the reward model scores $r(x, y)$ directly, for example, using PPO (Schulman et al., 2017). However, PPO-style methods would be more expensive, and instead, we formulate the task as a preference optimization problem so that LLM-generated programs can be *ranked* according to $r(x, y)$, which makes the objective amenable to preference-based RL algorithms such as (Rafailov et al., 2024; Calandriello et al., 2024). Preference optimization methods bypass the learning of the separate reward model and do not require a value function, which makes them more efficient than the on-policy RL methods. We provide a reinforcement learning formulation of `EvoTune` in Appendix A.5, along with a discussion of how it constitutes policy optimization in an off-policy manner.

For a preference data set $\mathcal{D}_{\mathrm{pref}} = \{(x^n, y_+^n, y_-^n)\}_{n=1}^N$, the loss function can be defined as the objective of direct preference optimization (DPO) (Rafailov et al., 2024; Wang et al., 2023):

$$\mathcal{L}(\pi_\theta; \pi_{\text{ref}}) = \mathbb{E}_{(x,y_+,y_-)\sim\mathcal{D}_{\text{pref}}}\left[-\log\sigma\left(\beta f'\left(\frac{\pi_\theta(y_+ \mid x)}{\pi_{\text{ref}}(y_+ \mid x)}\right) - \beta f'\left(\frac{\pi_\theta(y_- \mid x)}{\pi_{\text{ref}}(y_- \mid x)}\right)\right)\right]. \quad (3)$$

$\sigma$ represents the sigmoid function, and $f'$ depends on the chosen f-divergence. For example, for forward KL, $f'(u) = -1/u$, and for reverse KL, $f(u) = \log u + 1$.

## 3 EvoTune: Evolutionary search meets RL

EvoTune tightly combines evolutionary search with RL, where evolutionary search is used to discover new programs, and RL is subsequently used to optimize the policy with the better programs found by evolutionary search. Thus, EvoTune simultaneously improves both the outputs of the model and the model itself. We illustrate the pseudocode for EvoTune in Algorithm 1 and explain the two key components of our process: 1) Evolutionary search and 2) RL training.

### 3.1 Evolutionary search

In *evolutionary search* phase, EvoTune explores the space of possible programs to expand the program database denoted as $\mathcal{D}^t$. This database stores all valid programs generated up to the timestep $t$. At each timestep, EvoTune selects a subset of high-scoring programs from $\mathcal{D}^{t-1}$ and constructs a prompt $x^t$. The prompt is constructed by

---

**Algorithm 1** EvoTune

**Initialize:** Program database $\mathcal{D}^0$ with initial programs and policy (base LLM) $\pi_\theta^0$.
**for** $t = 1$ to $T$ **do**
  *Sample* a subset of programs from $\mathcal{D}^{t-1}$.
  *Construct* a new prompt $x^t$ from the sampled programs.
  *Generate K* outputs $\{y^{t,k}\}_{k=1}^K$ by sampling from $\pi_\theta^{t-1}(\cdot \mid x^t)$.
  *Extract* candidate programs from outputs and evaluate them on the validation set to obtain reward scores $r(y^{t,k})$.
  *Update* the program database:
    $\mathcal{D}^t \leftarrow \mathcal{D}^{t-1} \cup \{(x^t, y^{t,k}, r(y^{t,k}))\}_{k=1}^K$.
  **if** $t \bmod f_{\text{RL}} = 0$ **then**
    $\pi_\theta^t \leftarrow \text{RL-Update}(\pi_\theta^0, \mathcal{D}^t)$.
  **else**
    $\pi_\theta^t \leftarrow \pi_\theta^{t-1}$.
  **end if**
**end for**

---

concatenating $m = 2$ program-score pairs, followed by a task prompt that briefly describes the problem. This task prompt is structured in Chain-of-Thought (CoT) prompt style (Wei et al., 2022) (see Appendix A.7 for the prompt details) and encourages the LLM to identify patterns in how high-performing programs differ from the worse-performing ones.

The LLM generation policy $\pi_\theta^t$ is then conditioned on the prompt $x^t$ to generate $K$ new outputs $\{y^{t,k}\}_{k=1}^K$, where every output consists of a program and the rationale behind it. Each generated program is evaluated on a predefined set of validation task instances. Programs that are successfully evaluated without errors or exceeding computational constraints are assigned a score based on their performance. The evaluated programs and their respective scores are subsequently registered in the program database $\mathcal{D}^t$.

**Program database** Similar to the FunSearch method (Romera-Paredes et al., 2024), we use an island-based program database (Tanese, 1989; Cantú-Paz et al., 1998). We cluster programs into separate "islands" and evolve each island in isolation. Further details on the program database can be found in the Appendix A.4.

### 3.2 RL training

During the *RL training* phase, the generation policy $\pi_\theta^t$ is updated to improve its ability to generate high-quality outputs. After $f_{\text{RL}}$ search iterations, the policy is fine-tuned using RL objective on the accumulated dataset $\mathcal{D}^t$ to steer the policy toward generating programs with higher scores. While our framework is compatible with various RL algorithms in place of RL-Update(·) from Algorithm 1, we opt for DPO (Rafailov et al., 2024) due to its efficiency and simplicity.

**Preference dataset**   As DPO fine-tuning works with preferences, we update the preference dataset at each iteration $\mathcal{D}_{\text{pref}}^t = \mathcal{D}_{\text{pref}}^{t-1} \cup \{(x^t, y_+^{t,n}, y_-^{t,n})\}_{n=1}^{N^t}$. Each triplet consists of a prompt $x^t$ and two LLM outputs (each containing a program and a reasoning trace). The output containing the higher scoring program is denoted as $y_+^{t,n}$ and the output with the lower scoring program as $y_-^{t,n}$.

To construct triples $(x^t, y_+^{t,n}, y_-^{t,n})$, we start by taking all $K$ outputs $\{y^{t,k}\}_{k=1}^K$ generated from the same prompt $x^t$. The outputs with valid programs – those that successfully passed the evaluation – are divided into two groups according to their reward $r(y^{t,k})$: the higher-scoring and the lower-scoring half. We then randomly pair up members of these two groups so that each output $y^{t,k}$ is used at most once. In addition, we create extra preference pairs by matching failed outputs – those that contain an invalid program – with outputs containing a valid program. This process results in the design of $N^t$ preference pairs per prompt $x^t$.

To improve the quality of the dataset $\mathcal{D}_{\text{pref}}^t$, we employ an additional filtering step that excludes triplets $(x^t, y_+^{t,n}, y_-^{t,n})$ for which the reward of the higher scoring output $r(y_+^{t,n})$ does not exceed a dynamically determined threshold $\tau^t$. For a detailed description of threshold filtering, refer to Appendix A.6.

**Maintaining output diversity throughout training**   Output diversity is crucial for effective evolutionary search, yet RL fine-tuning can reduce it (Shumailov et al., 2024; Kirk et al., 2023; Casper et al., 2023). To mitigate this, we use a forward KL-regularized DPO objective (Equation 3), which encourages mass-covering behavior and avoids the mode collapse that is often induced by reverse KL (Wang et al., 2023). We set a high $\beta$ to ensure strong regularization. Additionally, we train on high-scoring programs from all search phases – not just recent ones – to maintain as much diversity as possible in the DPO dataset. Furthermore, each run is initialized from the base model $\pi_\theta^0$, following Singh et al. (2023). Hyperparameter details are in Appendix A.8.

## 4   Experiments

### 4.1   Evaluation tasks

We evaluate our approach on three well-known combinatorial optimization tasks that are suitable benchmarks for Python program generation by LLMs. Specifically, we focus on the online *bin packing* (BP) problem (Coffman Jr et al., 1984), the *traveling salesman* (TSP) problem (Jünger et al., 1995; Gutin & Punnen, 2006), and the *flatpack* (FP) problem (Bonnet et al., 2023).

**Bin packing**   In the BP problem, the objective is to assign an incoming stream of varying-sized items to as few fixed-size bins as possible. For each item, our method evolves a priority function (i.e., a Python program) that determines which bin should receive the item, given both the item's size and the current state of all bins (Romera-Paredes et al., 2024). To initialize the search, we begin with a best-fit heuristic function. This heuristic places each incoming item into the fullest bin with enough space to accommodate it.

**Traveling salesman problem**   With TSP, each problem instance is represented by a fully connected graph whose nodes correspond to cities and edges correspond to connections between them. Given a distance matrix specifying the inter-city distances, the objective is to find a minimal-distance route that passes through all cities. In our experiments, the evolved Python program is used in conjunction with a Guided Local Search (GLS) (Voudouris & Tsang, 1999; Alsheddy et al., 2018), following previous work by Liu et al. (2024); Ye et al. (2024). The LLM's task is to propose heuristics for computing a penalty matrix based on the input distance matrix. This penalty is used iteratively in the GLS procedure to penalize certain edges in the solution. For TSP, we use the identity function as the starting point for evolutionary search.

**Flatpack**    For FP, each problem instance is represented by a two-dimensional grid and a set of randomly generated connected blocks of maximum size $3 \times 3$. The objective is to sequentially place all blocks in any rotation onto the grid without overlap, maximizing the fraction of the grid covered. In our experiments, the evolved Python program receives the current state of the grid as input, and outputs scores for each possible combination of block, rotation, and placement location. Higher scores are interpreted as better placements, and the combination with the highest score that results in a valid placement is selected. This process is repeated sequentially until no more blocks can be placed. We initialize the search with a function that assigns the same score all all possible block, rotation, and location combinations.

**Evaluation protocol**    In addition to evaluating our method on the *validation* set, we also evaluate on the *validation-perturbed* set with controlled modifications of the validation set, and the *test* set consisting of new problem instances drawn from the same distribution as the validation set. For all tasks, performance is measured with the optimality gap. Details on the dataset construction, instance perturbation, and metric definitions are provided in Appendix A.1.

**Broader Evaluation Scope**    To further evaluate the generality of our approach, we include two additional sets of benchmarks. First, we tackle two real-world combinatorial optimization challenges from the **Google Hash Code programming competition** (Veličković et al., 2024), which involve optimally placing servers in a datacenter to maximize fault tolerance and assigning a fleet of self-driving cars to ride requests. Second, we address two **symbolic regression tasks** from the LLM-SR benchmark suite (Shojaee et al., 2024). These tasks aim to uncover underlying scientific equations from observational data and require modeling the mechanical stress behavior of a material and discovering a differential equation for bacterial growth dynamics. Appendix A.1 provides detailed descriptions of these additional tasks.

## 4.2   Results

We compare `EvoTune` against a FunSearch-style baseline (Romera-Paredes et al., 2024) denotes simply as FunSearch. This baseline uses only evolutionary search and does not involve training the LLM. We test our method on three instruction-tuned LLMs: Llama3.2 1B Instruct (Dubey et al., 2024), Phi 3.5 Mini Instruct (Abdin et al., 2024), and Granite 3.1 2B Instruct (Granite Team, 2024). To account for the high experimental variability, each reported result is the average of ten random seeds. Our objective is to maximize the reward, which we define as the negative of the optimality gap. Hence, minimizing the optimality gap is equivalent to maximizing the reward.

During program evolution ($t = 0, \ldots, T$), we evaluate LLM-generated programs on a problem-specific *validation* set of problem instances. In Figure 2 (*Top*) we report the progression of the reward of the 50 best-performing discovered programs. Compared to evaluating a single best program, the top-50 metric allows us to obtain more robust estimates as it indicates whether the search policy found more promising regions within the search space, rather than sampling an isolated high-reward program by chance. For completeness, we also report the best overall program scores (top 1 scores) in the Appendix A.10.

Across all evaluated LLMs and problem domains, our method, `EvoTune`, consistently achieves higher final top-50 reward scores compared to the baseline. This improvement demonstrates that refining the search policy via RL accelerates the discovery of high-quality algorithms. Notably, in most cases, the performance gap between the baseline and `EvoTune` widens as more programs are sampled, suggesting that larger sampling budgets could amplify our method's advantage over the baseline.

In addition to the top-50 metric, `EvoTune` attains higher average reward scores across all generated programs relative to the baseline. We note that nearly every training run resulted in an increased average reward. However, achieving significant gains in the performance of the top programs (top 50 and top 1) required more careful tuning. Results in terms of average rewards are detailed in Figure 5 in Appendix A.10.

| | VALIDATION SET | | | VALIDATION-PERTURBED SET | | | TEST SET | | |
|---|---|---|---|---|---|---|---|---|---|
| | 9.6K | 16K | 22.4K | 9.6K | 16K | 22.4K | 9.6K | 16K | 22.4K |
| **BIN PACKING** | | | | | | | | | |
| LLAMA FUNSEARCH | 5.08 ± 0.09 | 4.67 ± 0.15 | 4.35 ± 0.16 | 4.46 ± 0.14 | 4.01 ± 0.19 | 3.68 ± 0.18 | 4.52 ± 0.15 | 4.07 ± 0.19 | 3.77 ± 0.18 |
| LLAMA EVOTUNE | 4.96 ± 0.09 | 4.11 ± 0.17 | 3.73 ± 0.15 | 4.31 ± 0.15 | 3.39 ± 0.20 | 3.10 ± 0.17 | 4.39 ± 0.15 | 3.51 ± 0.19 | 3.21 ± 0.14 |
| PHI FUNSEARCH | 4.47 ± 0.13 | 3.86 ± 0.12 | 3.60 ± 0.10 | 3.89 ± 0.18 | 3.34 ± 0.14 | 3.09 ± 0.11 | 3.99 ± 0.17 | 3.42 ± 0.13 | 3.19 ± 0.09 |
| PHI EVOTUNE | 3.81 ± 0.12 | 3.40 ± 0.08 | 3.12 ± 0.07 | 3.31 ± 0.17 | 2.88 ± 0.11 | 2.70 ± 0.01 | 3.38 ± 0.17 | 2.99 ± 0.12 | 2.80 ± 0.10 |
| GRANITE FUNSEARCH | 3.64 ± 0.08 | 3.42 ± 0.05 | 3.33 ± 0.06 | 3.12 ± 0.09 | 2.95 ± 0.07 | 2.86 ± 0.08 | 3.20 ± 0.07 | 3.05 ± 0.06 | 2.97 ± 0.08 |
| GRANITE EVOTUNE | 3.50 ± 0.10 | 3.32 ± 0.08 | 3.18 ± 0.05 | 2.92 ± 0.14 | 2.75 ± 0.08 | 2.66 ± 0.07 | 3.07 ± 0.12 | 2.89 ± 0.07 | 2.82 ± 0.07 |
| **TRAVELING SALESMAN PROBLEM** | | | | | | | | | |
| LLAMA FUNSEACH | 2.591 ± 0.003 | 2.575 ± 0.003 | 2.565 ± 0.004 | 2.937 ± 0.002 | 2.929 ± 0.002 | 2.922 ± 0.002 | 2.594 ± 0.002 | 2.580 ± 0.004 | 2.572 ± 0.004 |
| LLAMA EVOTUNE | 2.580 ± 0.002 | 2.564 ± 0.003 | 2.554 ± 0.003 | 2.928 ± 0.002 | 2.918 ± 0.002 | 2.912 ± 0.002 | 2.582 ± 0.003 | 2.573 ± 0.001 | 2.566 ± 0.002 |
| PHI FUNSEARCH | 2.610 ± 0.003 | 2.593 ± 0.003 | 2.583 ± 0.003 | 2.950 ± 0.002 | 2.941 ± 0.001 | 2.936 ± 0.001 | 2.647 ± 0.005 | 2.624 ± 0.006 | 2.611 ± 0.004 |
| PHI EVOTUNE | 2.589 ± 0.005 | 2.567 ± 0.005 | 2.551 ± 0.005 | 2.942 ± 0.001 | 2.931 ± 0.002 | 2.921 ± 0.003 | 2.617 ± 0.007 | 2.592 ± 0.006 | 2.575 ± 0.006 |
| GRANITE FUNSEARCH | 2.565 ± 0.005 | 2.545 ± 0.005 | 2.534 ± 0.006 | 2.933 ± 0.004 | 2.921 ± 0.004 | 2.911 ± 0.005 | 2.575 ± 0.004 | 2.559 ± 0.005 | 2.548 ± 0.005 |
| GRANITE EVOTUNE | 2.546 ± 0.004 | 2.521 ± 0.004 | 2.504 ± 0.004 | 2.921 ± 0.003 | 2.905 ± 0.004 | 2.894 ± 0.004 | 2.565 ± 0.003 | 2.546 ± 0.003 | 2.534 ± 0.003 |
| **FLAT PACK** | | | | | | | | | |
| LLAMA FUNSEARCH | 0.168 ± 0.002 | 0.155 ± 0.004 | 0.148 ± 0.004 | 0.154 ± 0.003 | 0.141 ± 0.006 | 0.135 ± 0.006 | 0.165 ± 0.004 | 0.152 ± 0.005 | 0.148 ± 0.005 |
| LLAMA EVOTUNE | 0.150 ± 0.003 | 0.136 ± 0.003 | 0.126 ± 0.004 | 0.138 ± 0.004 | 0.122 ± 0.004 | 0.112 ± 0.004 | 0.149 ± 0.002 | 0.134 ± 0.003 | 0.126 ± 0.003 |
| PHI FUNSEARCH | 0.163 ± 0.004 | 0.137 ± 0.005 | 0.125 ± 0.004 | 0.154 ± 0.006 | 0.124 ± 0.005 | 0.111 ± 0.003 | 0.166 ± 0.005 | 0.142 ± 0.005 | 0.131 ± 0.003 |
| PHI EVOTUNE | 0.156 ± 0.008 | 0.127 ± 0.006 | 0.115 ± 0.003 | 0.139 ± 0.007 | 0.116 ± 0.006 | 0.106 ± 0.002 | 0.156 ± 0.006 | 0.133 ± 0.006 | 0.121 ± 0.003 |
| GRANITE FUNSEARCH | 0.117 ± 0.001 | 0.113 ± 0.001 | 0.111 ± 0.001 | 0.105 ± 0.001 | 0.103 ± 0.001 | 0.101 ± 0.001 | 0.124 ± 0.001 | 0.120 ± 0.001 | 0.118 ± 0.001 |
| GRANITE EVOTUNE | 0.113 ± 0.001 | 0.109 ± 0.000 | 0.105 ± 0.001 | 0.103 ± 0.001 | 0.101 ± 0.001 | 0.099 ± 0.001 | 0.120 ± 0.000 | 0.116 ± 0.000 | 0.113 ± 0.001 |

Table 1: **Results for Bin Packing *(Top)*, Traveling Salesman Problem *(Middle)*, and Flatpack *(Bottom)*.** We report mean *optimality gaps of top 50 programs* and standard error across 10 seeds on validation, validation-perturbed, and test sets at three different sampling budgets (9.6k, 16k and 22.4k sampled programs, corresponding to the x-axis in Figure 2). Across different models, tasks, and sampling budgets, EvoTune consistently outperforms FunSearch. The best performance is highlighted in  blue .

In many mathematical problems, the challenge lies in finding optimal solutions within a specific search space, regardless of their generalization beyond it. For instance, many problems require identifying high-quality solutions within particular dimensions, where cross-dimensional generalization is not the primary concern (Grochow, 2019; MacWilliams & Sloane, 1977). The observed improvement in search performance on the validation set using EvoTune is thus a promising indicator of its potential to address such challenging mathematical problems.

To further assess the robustness and generalization of the generated programs, we evaluate their performance on the validation-perturbed and test sets. For these evaluations, we

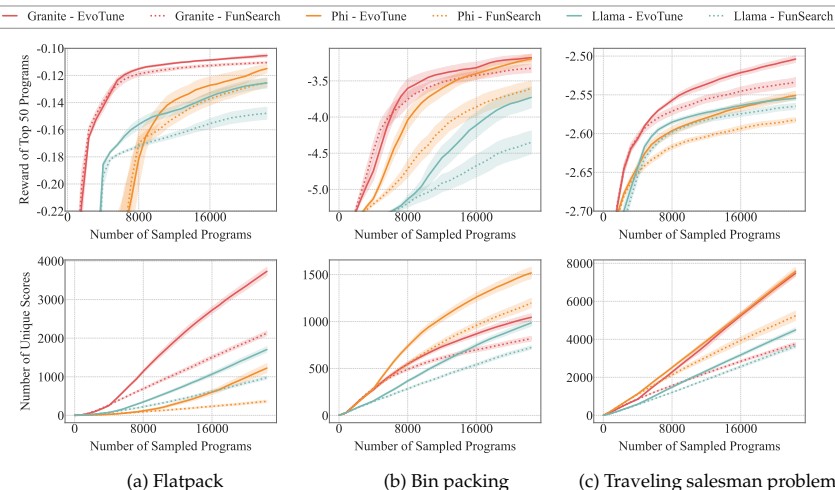

(a) Flatpack  (b) Bin packing  (c) Traveling salesman problem

Figure 2: **Top-50 rewards and the number of unique scores.** The reward score of the best 50 generated programs (*Top*) and the number of programs with unique scores across different models *Bottom* for (a) flatpack, (b) bin packing, and (c) traveling salesman problem. The shaded areas denote the standard error computed over 10 seeds. Across all models and tasks, EvoTune finds higher-scoring best 50 programs. Additionally, it finds a greater number of uniquely scoring solutions.

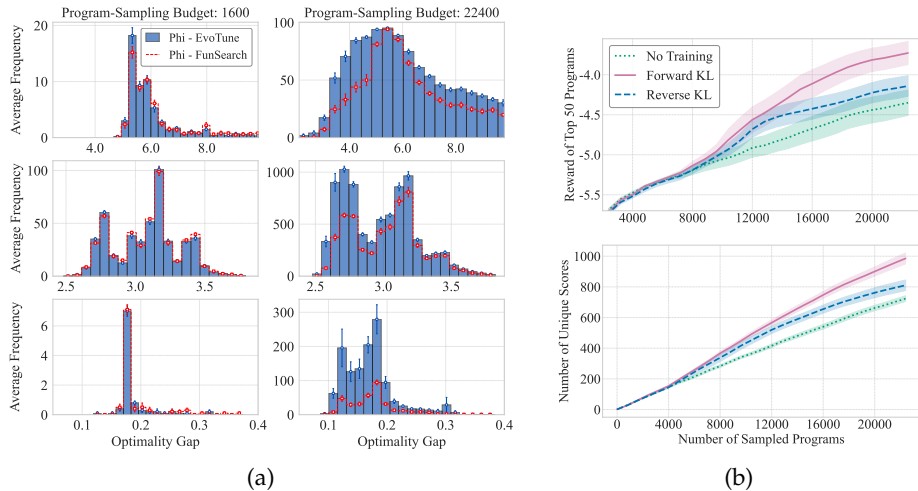

(a)                           (b)

Figure 3: **(a) Evolution of optimality gap distributions**. Histograms illustrating the distribution of optimality gap scores for programs in the program database at an early checkpoint with limited sampling budget *(Left)* and at the final checkpoint with full sampling budget *(Right)*. The *Top*, *Middle*, and *Bottom* rows show results for the BP, TSP, and FP tasks, respectively. All results are averaged over 10 seeds. Throughout the search process, `EvoTune` produces a higher number of high-quality programs (indicated by lower optimality gap scores) compared to the baseline. **(b) Forward KL vs. Reverse KL**. Comparison of KL variants based on the reward of the top 50 programs *(Top)* and the number of unique scores *(Bottom)*. Forward KL yields higher rewards and a higher number of unique solutions, which we attribute to a higher diversity of outputs.

consider all programs in the program database that are generated up to a given sampling budget and measure their performance on the corresponding evaluation sets. As shown in Table 1, our method outperforms the baseline on both the validation-perturbed and test set across all tasks and models.

**Number of unique solutions discovered** Figure 2 (*Bottom*) illustrates the number of unique solutions found by the methods, as measured by the number of unique evaluation scores in the program database. Although achieving a higher count of unique solutions is not strictly necessary for finding higher scoring programs, it indicates a more comprehensive search uncovering a wider range of potential solutions. Across all benchmarks and LLMs, `EvoTune` consistently discovers a greater number of unique solutions compared to the baseline. For any specific model and benchmark, achieving a higher number of unique solutions correlates with achieving higher rewards. Additionally, while tuning the training hyperparameters, tracking the unique solution metric proved effective in signaling when the training began to overfit.

**Distribution of scores in program database** Figure 3a compares how the distribution of scores within the program database evolves from an early-stage checkpoint to the final one for both methods. Initially, the distributions of the optimality gap scores of both methods are comparable. As the search progresses, `EvoTune` exhibits a greater increase in the frequency of high-scoring solutions relative to the baseline. While the counts across all optimality gap scores increase more with `EvoTune`, this increase is especially pronounced in the highest-quality region (i.e., low optimality gap). This improves the chances of finding record-breaking mutations beyond the current frontier, as more diverse candidate solutions become available to the model to innovate upon.

We include similar optimality gap distributions for all models and benchmarks in Appendix A.10, along with a complementary t-SNE analysis of function embeddings (Figure 9a, Figure 9b) that provides structural insight beyond score-based diversity.

**Forward vs. reverse KL** As discussed in Section 3.2, a key design choice in our RL phase is using the *forward* KL variant of DPO rather than the more commonly used *reverse* KL

(Rafailov et al., 2024). To evaluate the impact of this choice, we conducted an ablation study on the bin packing task using the Llama3.2 1B Instruct model, comparing the performance of forward and reverse KL regularization. As depicted in Figure 3b *(Top)*, both KL variants of our method surpass the FunSearch baseline, but the forward KL variant discovers the best-performing programs. In addition, it generates a greater number of unique solutions, as shown in Figure 3b *(Bottom)*, demonstrating its effectiveness in promoting output diversity.

Additionally, we evaluated an alternative RL algorithm: the ReST$^{EM}$ approach (Singh et al., 2023; Gulcehre et al., 2023). This offline RL method iteratively applies supervised fine-tuning (SFT) on high-scoring outputs. Our experiments indicate that ReST$^{EM}$ underperforms relative to DPO and is more sensitive to hyperparameters. Detailed results for this investigation are provided in the Appendix A.13.

While we improve the baseline method by adding *in-weight* learning to the LLM, we also tried improving *in-context learning* (Dong et al., 2022) by providing more examples in the prompt, but could not gain considerable improvements.

**Results on Hash Code and LLM-SR problems**   To expand the scope and further test applicability and generality of our method, we benchmark it on two problems from the Hash Code programming competition organized by Google and two problems from the LLM-SR benchmark suite on discovering scientific equations.

The results, shown in Figure 4, are consistent with our previous findings - EvoTune consistently outperforms FunSearch in terms of both the reward of the best 50 solutions found and the number of unique solutions discovered during the evolutionary process.

Notably, on the Datacenter Optimization task from Hashcode, EvoTune achieves a score of 418 surpassing the competition's top human score of 407. FunSearch also improves over the human baseline, achieving a score of 414, but it does not match Evotune's peak performance. For this task, we set the sampling budget for both methods at 10,000 functions. Across the LLM-SR tasks, EvoTune consistently outperforms the evolutionary search baseline, both throughout the optimization trajectory and on the final in-distribution (ID) and out-of-distribution (OOD) evaluation sets. On the Stress-Strain task, our method, even when using the small Phi-3.5 Mini (3.8B) model, surpasses baselines that rely on larger or proprietary models such as Mixtral 8x7B and GPT-3.5-turbo.

These results demonstrate that our method scales effectively to real-world settings and discovers high-quality solutions for these problems. Full results for these tasks can be found in Appendix A.9

**Comparison to non-LLM baselines**   To contextualize EvoTune's performance, we benchmarked it against specialized non-LLM approaches. Our evaluation shows that this general-purpose method can discover solutions that surpass both task-specific methods and established human-designed heuristics. Full experimental details are available in Appendix A.12.

## 5   Related Work

**Evolutionary search with LLMs**   Lehman et al. (2023) introduce ELM for evolving Python programs that configure walking robots, using RL *only* to condition generation in new domains – not to improve the search itself. In contrast, we integrate RL with evolutionary search to refine the generator policy. Liu et al. (2024) evolve heuristics and their *thoughts* with an LLM without training. Ye et al. (2024) extend this with a self-reflecting LLM and specialized evolutionary steps. Liu et al. (2023a) evolve optimization algorithms via prompt-based mutation and crossover without reward feedback. Liu et al. (2023b) propose LMEA, which relies on carefully curated prompts and directly evolves solutions in natural language, not algorithms, which does not scale with the size of the problem.

**Prompt optimization**   A closely related line of research explores how to vary and optimize the prompts to better elicit desirable outputs from LLMs. Yang et al. (2024) leverage the LLM

as a prompt optimizer that directly outputs solutions as a black-box method without iterating over algorithms and without updating the generator policy. Guo et al. (2023) introduce EvoPrompt, which integrates evolutionary search with LLMs for prompt optimization without a learning component. Similarly, Fernando et al. (2023) developed an evolutionary method that self-referentially evolves and improves the prompts and mutation operators jointly.

**Self-improvement and self-training**    Iterative self-improvement training techniques work by training models using their own generated outputs (Zelikman et al., 2022; Gulcehre et al., 2023; Singh et al., 2023; Ishibashi et al., 2024; Pang et al., 2024). Candidate solutions are generated and then filtered based on correctness or alignment with predefined criteria. The selected outputs are used to fine-tune the model, and this cycle is repeated, gradually improving its ability to produce desirable solutions. Our method adds to the repertoire of self-improving techniques – by training the LLM on self-discovered solutions, we improve the evolutionary search capabilities of the model. For a more comprehensive discussion of additional related work, including neural combinatorial optimization, we refer the reader to Appendix A.3.

## 6    Conclusion

We found that existing evolutionary search approaches can converge to suboptimal solutions with a limited sampling budget. To address this, we introduced `EvoTune`, a novel approach that integrates evolutionary search with RL fine-tuning to improve LLM-driven algorithm discovery. By iteratively refining the LLM through RL finetuning, our method outperforms a purely search-based baseline on challenging combinatorial optimization and symbolic regression tasks Our results establish the viability of integrating RL into evolutionary search such that (i) the training effectively guides the evolutionary search toward superior solutions (measured by the single best or top-k performance) rather than merely increasing the average score of the population and (ii) the diversity of sampled functions is maintained, a critical and non-trivial challenge in self-improvement training, which we achieve through techniques like Forward KL regularization.

Although our results highlight the promise of `EvoTune`, several questions remain for further investigation. Our experiments were conducted with LLMs ranging from 1B to 3.8B parameters and with a sampling budget of up to 22.4k outputs. Further scaling of both the model size and sampling budget is needed to fully understand the method's potential. Furthermore, while `EvoTune` discovers better solutions within a fixed sampling budget, it incurs additional compute costs due to the RL training phase. Investigating the trade-offs between training and inference costs, especially at larger scales, is an important direction for future research.

In a nutshell, `EvoTune` demonstrates the potential of combining the synergistic strengths of evolutionary search and reinforcement learning, paving the way for future advances in LLM-based algorithm discovery.

## Ethics Statement

LLMs are dual-use technologies capable of serving both beneficial and potentially harmful purposes. Enhancing their capabilities, as demonstrated in this work, can advance the discovery of automated algorithms, fostering innovations in various scientific and industrial domains. However, these advancements raise concerns about misuse, such as generating malicious algorithms. It is crucial to implement robust safeguards and ethical guidelines to mitigate these risks, ensuring that the improved capabilities of LLMs are harnessed responsibly and for the greater good of society.

## Acknowledgements

We are grateful to Bernardino Romera Paredes and Alhussein Fawzi for the insightful discussions that contributed to this work. We also thank the SwissAI Initiative and the SCITAS team at EPFL for providing the computational resources that enabled our research. We extend our appreciation to Karin Getaz for administrative support, and to Skander Moalla and Yugesh Ajit Kothari for their assistance with the technical implementation.

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

# A  Appendix

## A.1  Evaluation tasks

**Bin packing problem**  BP validation set consists of 20 packing instances, each containing 500 items sampled according to the OR-Library (Beasley, 1990). Performance is measured as the fraction of excess bins used in the lower bound (Martello & Toth, 1990). To generate the validation-perturbed dataset, we randomly perturb the order of items in the validation dataset to measure the robustness of programs when presented with perturbed but familiar inputs. To generate the test set, we sample new problem instances from the same distribution as the OR dataset (Romera-Paredes et al., 2024).

**Traveling salesman problem**    For a TSP instance of size $c$, we sample $c$ pairs of $(x, y)$ city coordinates uniformly from $[0, 1]^2$, and use the distance matrix as input to the GLS procedure (see Appendix A.2 for more details on GLS). Performance is measured as the fraction of excess cost incurred by the calculated route over the optimal route given by the Elkai solver (Dimitrovski, 2019). The validation set consists of 100 problem instances of size $c = 100$ and 100 instances of size $c = 200$. To generate the validation-perturbed set, we alter the adjacency matrix such that the cost of each edge is replaced by a high value with probability $p = 0.2$. This allows us to evaluate the robustness of the generated programs in proposing heuristics that work well when the input is slightly changed. It resembles a real-world situation where the connection between two cities is suddenly cut off or travel is slow. To generate the test set, we sample a new batch of TSP instances using the same procedure as for the validation set.

**Flatpack problem**    The validation set consists of 45 problem instances, with 15 instances using a $9 \times 9$ grid, 20 instances using an $11 \times 11$ grid, and 10 instances using a $15 \times 15$ grid. The validation-perturbed set is constructed by modifying each instance in the training set: A rectangle of size $\lfloor \sqrt{r} + 0.5 \rfloor \times \lfloor \sqrt{c} + 0.5 \rfloor$ is placed in the center of the grid, where $r$ and $c$ denote the number of rows and columns of the grid, respectively. This obstacle prevents block placements in the center region and allows us to evaluate the robustness of generated heuristics when confronted with partially obstructed configurations. The test set follows the same grid size distribution as the training set.

**Hash Code datacenter optimization:**    This problem, taken from the 2015 Google Hash Code qualification round, involves optimally placing servers of varying sizes and capacities into a grid-like data center while assigning them to logical pools to maximize fault tolerance. The data center consists of rows and slots, some of which may be unavailable, and each server must be placed in a contiguous sequence of unblocked slots and assigned to a pool. The goal is to maximize the guaranteed capacity, defined as the minimum remaining capacity in any pool if a single row fails. Formally, this is a min–max optimization problem: for each pool, the guaranteed capacity equals the total capacity of its servers minus the maximum capacity loss from a single row failure, and the overall objective is to maximize the minimum such value across all pools. Analogous to Veličković et al. (2024), we evolve a heuristic function `score_greedy()`, which evaluates server placements and pool assignments in two distinct phases. This function takes as input a server, a candidate row, an optional pool, and the current distribution of capacities, and outputs a score reflecting the desirability of the configuration. The initial heuristic function prioritizes servers with high capacity-to-size ratios for placement, while discouraging assignments that disproportionately concentrate capacity within a single row of a given pool. All functions are evaluated on a fixed instance from the original competition benchmark.

**Hash Code self-driving rides**    This problem, taken from the 2018 Google Hash Code qualification round, involves assigning a fleet of self-driving cars to a set of ride requests on a grid. Each ride order is constrained by a start and end location point, earliest start time, and a latest finish time. Following the setup from Veličković et al. (2024), our approach evolves a function, `pick_rides()`, which takes as input a car's current location and time, along with a tuple of candidate rides, and returns the index of the selected ride. The greedy initial policy simply selects the first feasible ride, if it exists. Performance is evaluated on holdout datasets from the competition, with scores based on the total distance covered and bonuses for early starts.

**LLM-SR material stress behavior**    This task models the mechanical behavior of Aluminium 6061-T651 under tension across six temperature settings (20°C to 300°C), based on real experimental data. The goal is to predict stress as a function of strain and temperature. Unlike classical physics problems with known governing equations, this setting lacks a standard closed-form solution due to complex, nonlinear, and piecewise behavior in the material's response across different conditions. These characteristics make the task particularly challenging for symbolic regression, as it requires uncovering empirical patterns without relying on established physical laws. The regression search begins from a simple linear

model, with parameters optimized via gradient descent. To assess generalization, data at 200°C is held out as an out-of-domain (OOD) validation set.

**LLM-SR bacterial growth modeling**    This task involves modeling the growth rate dynamics of Escherichia coli (E. coli) bacteria using a differential equation that accounts for key environmental and biological factors such as population density, substrate concentration, temperature, and pH level. The benchmark reflects the prior biological knowledge; however, to reduce the possibility of models solving the task through memorization, the task uses synthetic data generated from custom formulations, rather than standard textbook equations. This encourages reasoning and exploration over mere recall. Out-of-domain (OOD) set includes parameters outside the ranges seen during the equation evolution, thereby testing generalization to unfamiliar conditions. The search is initialized from a simple multiplicative initial equation the evaluation score is mean squared error (MSE).

## A.2   Guided local search

Guided Local Search (GLS) (Voudouris & Tsang, 1999; Alsheddy et al., 2018) is an optimization technique designed to improve the performance of local search algorithms by helping them escape local optima. This is achieved by penalizing certain feature sets of solutions that contribute to suboptimal solutions, thereby guiding the search process to more promising areas of the solution space. GLS works by iteratively adjusting the objective function with a penalty term, discouraging the search from revisiting or remaining in areas of the search space that contain undesirable features.

When applied to the Traveling Salesman Problem (TSP), GLS can improve the efficiency of local search methods. In the context of TSP, GLS penalizes edges (or tours) that frequently appear in suboptimal solutions, thus encouraging the search to explore alternative routes. By systematically guiding the local search away from suboptimal solutions, GLS helps in finding shorter and more optimal tours. Similar to Ye et al. (2024), we use a variation of GLS that interleaves local search with perturbations (Arnold & Sörensen, 2019). More specifically, at the beginning of the GLS procedure, an initial tour is obtained using the nearest neighbor heuristic, then for $i$ rounds, we alternate between local search and perturbation, updating the best tour ($i = 16, 8$ for $c = 100, 200$, respectively).

Local search consists of two operations: a) Relocate-Once, b) Two-Opt. The Relocate-Once operation involves removing a single city from its current position in the tour and inserting it into a different position. This move aims to explore the impact of shifting from one city to another location in the tour, potentially leading to a shorter overall path. The Two-Opt operation is a well-known heuristic that involves selecting two edges in the tour, removing them, and reconnecting the segments in a different way that still results in a valid tour. This operation can effectively eliminate crossings in the tour, which are often associated with suboptimal solutions, leading to a shorter and more efficient route.

The perturbation operation in the GLS uses controlled disruptions to the current solution to escape local optima, leveraging a guide computed using programs generated by the LLM that take the distance matrix as input. The perturbation operation iteratively penalizes certain edges in the current tour, encouraging the search to explore alternative routes. The goal is to modify the solution such that it escapes local optima and continues searching for a global optimum. For each edge in the current tour, a utility value is computed using the guide provided by the LLM. The edge with the highest utility value is identified as the most promising candidate for perturbation and the penalty associated with the selected edge is incremented, discouraging the local search from selecting this edge in subsequent iterations. This helps diversify the search space by effectively increasing the cost of returning to previously explored (and penalized) solutions. This is followed by the distance matrix being adjusted by the weighted penalties, resulting in a new guided edge weight matrix which directs the subsequent local search by reflecting both the original distances and the imposed penalties.

### A.3 Extended related work

**Neural combinatorial optimization (NCO)** NCO is an important orthogonal class of methods that learn to construct solutions for combinatorial optimization problems using embeddings of problem instances as input. Such models can be trained with supervised learning (Vinyals et al., 2015; Joshi et al., 2019; Fu et al., 2021; Joshi et al., 2022; Kool et al., 2022; Hottung et al., 2021a) or RL (Bello et al., 2016; Kool et al., 2018; Hottung & Tierney, 2020; Hottung et al., 2021b; Chen & Tian, 2019) (See Luo et al. (2023) for more references). Our work on the other hand does not deal with problem instances directly; rather, it searches for an algorithm or heuristic that can later be utilized with problems of any size.

### A.4 Program database

Inspired by Romera-Paredes et al. (2024), we organize our program database into islands and clusters (Tanese, 1989; Cantú-Paz et al., 1998). Each island represents a group of programs that evolve independently. Within an island, programs are further grouped into clusters based on their scores.

We use the following procedure to form the prompt $x^t$, which consists of $m$ programs sampled from the program database. First, we select an island $i$ uniformly at random. Next, we sample $m$ clusters from the chosen island $i$. This ensures that the selected programs, which will be used to construct the prompt, have different scores, making it possible for the LLM to identify "the direction" of improvement. Additionally, we observed that programs within the same cluster often differ only superficially (e.g., minor variations in subroutines or variable names while performing the same computation). Hence, to avoid constructing prompts with overly similar programs, we sample different clusters. To sample $m$ clusters, we draw from a softmax distribution over cluster scores, using a temperature parameter. We also incorporate an annealing strategy (Kirkpatrick et al., 1983) to adjust sampling over time such that toward the later stages, the clusters will be sampled from the top $p_t > p_{t-1}$ percentile of the database to balance exploration-exploitation. After choosing $m$ clusters, we sample one program from each cluster, prioritizing shorter programs (Romera-Paredes et al., 2024). Unlike Romera-Paredes et al. (2024), we do not reset the islands that contain low-scoring programs.

After sampling $m$ programs from an island $i$, we construct the prompt $x^t$ as detailed in Appendix A.7. All newly generated outputs from this prompt are placed back in the same island $i$. This approach helps prevent excessive similarity between programs in the database and encourages the exploration of a wider set of ideas.

### A.5 RL formulation of `EvoTune`

In this section, we detail the reinforcement learning framework underlying `EvoTune` and explain why we classify it as a reinforcement learning approach, although its optimization is performed using DPO.

We define an MDP $\mathcal{M} = (\mathcal{S}, \mathcal{A}, \mathcal{R}, \mathcal{T})$ as follows:

- **States** ($\mathcal{S}$): Partial sequences $(x, y_{1:k})$ consisting of prompt $x$ and a partially generated output of length $k$. Note that the prompts $x$ are not fixed, unlike the formulation in the DPO paper.

- **Actions** ($\mathcal{A}$): Sampling the next token $y_{k+1} \in \mathcal{V}$ from the vocabulary $\mathcal{V}$.

- **Rewards** ($\mathcal{R}$): Rewards assigned by the Bradley-Terry reward model in terminal state $K$ (end of sequence token or full context). The reward reflects the performance on the task-specific evaluation set over the generated program extracted from the output $y_{1:K}$. The reward reflects the quality of the generated program and is undiscounted, meaning the discount factor is set to 1.

- **Transitions** ($\mathcal{T}$): Deterministic transitions from $(x, y_{1:k})$ to $(x, y_{1:k+1})$ as new tokens are appended to the sequence.

The RL problem in the context of preference learning could be formulated as:

$$\arg\max_{\theta} \; \mathbb{E}_{\substack{x \sim \mathcal{D}^t \\ (y_+, y_-) \sim \pi_\theta}} \left[ p(y_+ \succ y_- | x) - \beta \mathbb{D}_f(\pi_{ref}(\cdot|x) || \pi_\theta(\cdot|x)) \right]. \tag{4}$$

**Off-policy DPO training**  DPO was proposed as an RL-free method, as the reward model it optimizes is implicit, and training can be conducted entirely offline. However, it still performs constrained reward maximization but by adopting the Bradley–Terry reward model $p(y_+ \succ y_- | x)$, it diverges from the classical RLHF training setting. While standard DPO operates on fixed offline samples, `EvoTune` functions in an off-policy setting, as the updated model is iteratively used to generate new outputs – enabling further performance improvements. In this sense, our method is similar to iterative DPO (Pang et al., 2024; Ishibashi et al., 2024). Furthermore, the samples in the program database [1] are dynamically generated and not fixed. We believe that this makes the DPO approach in `EvoTune` closer to a more traditional RL algorithm.

**Alternative RL formulation**  The RL problem that DPO optimizes can also be seen as a one-step offline RL problem (Gülçehre et al., 2020; Levine et al., 2020) or an offline *bandit* (Lattimore & Szepesvári, 2020). Since rewards are only provided at terminal states and transitions are fully deterministic, the problem reduces to a bandit formulation, where an action corresponds to sampling the entire output $y_{1:K}$.

## A.6 DPO dataset filtering

To reduce training time and computational cost, we apply a filtering procedure to reduce the size of the DPO dataset $\mathcal{D}^t_{\text{pref}}$. This ensures that the training focuses on high-quality data points while maintaining diversity. We filter out any datapoints $(x^t, y^{t,n}_+, y^{t,n}_-)$ where the reward of the higher scoring output $y^{t,n}_+$ falls under a predefined threshold $\tau^t$. The threshold $\tau^t$ is calculated based on the distribution of rewards from the outputs generated since the last *RL training* phase. More specifically, it is set as the 30th percentile of rewards from newly generated outputs. As the average reward improves over time, this threshold also naturally improves, ensuring that only progressively better outputs are retained.

## A.7 LLM prompts

We present here the system prompts as well as the task descriptions for BP, TSP, and FP in Prompt 1, 2, 3, and 4. A complete query to the LLM consists of concatenating the system prompt, two sampled programs accompanied by their score, and the task description.

```
You are a helpful, excellent, and innovative problem-solver specializing in mathematical
optimization and algorithm design.  You are an expert in writing Python functions.
```

Prompt 1: System prompt for the LLM.

```
You are tasked with creating a new function, priority(), that outperforms the other two
presented functions.
To achieve this, follow these guidelines:
Think Outside the Box:  Avoid simply rewriting or rephrasing existing approaches.
Prioritize creating novel solutions rather than making superficial tweaks.
Analyze the Score Drivers:  Analyze the characteristics of the higher-scoring function.
Identify what it is doing differently or more effectively than the lower-scoring function.
Determine which specific changes or techniques lead to better performance.
Experiment with Variations:  Use the insights to create a new function that builds upon
successful ideas but introduces innovative variations.  Consider entirely new strategies
or optimizations that were not present in the previous attempts.
To summarize, your task is to write a new function named priority() that will perform
better than both functions above and achieve a higher score.
```

---

[1] Let us note that that the program database is similar to replay buffer in off-policy RL (Mnih et al., 2015; Schaul et al., 2015)

Prompt 2: Description of the bin packing problem.

```
You are tasked with creating a new function, heuristics(), that outperforms the other two
presented functions.
The heuristics() function takes as input a distance matrix, and returns prior indicators
of how undesirable it is to include each edge in a solution.  The returned matrix should
be of the same shape as the input.
When writing the new function, follow these guidelines:
Think Outside the Box:  Avoid simply rewriting or rephrasing existing approaches.
Prioritize creating novel solutions rather than making superficial tweaks.
Analyze the Score Drivers:  Analyze the characteristics of the higher-scoring function.
Identify what it is doing differently or more effectively than the lower-scoring function.
Determine which specific changes or techniques lead to better performance.
To summarize, your task is to write a new function named heuristics() that will perform
better than both functions above and achieve a higher score.
```

Prompt 3: Description of the traveling salesman problem.

```
You are tasked with creating a new function, priority(), that outperforms the other two
presented functions.
The priority() function takes three inputs:
1.  current_grid:  numpy array (float32) of shape (num_rows, num_cols) with values in the
range [0, num_blocks] (corresponding to the number of each block).  This grid will have
zeros where no blocks have been placed and numbers corresponding to each block where that
particular block has been placed.
2.  blocks:  numpy array (float32) of shape (num_blocks, 3, 3) of all possible blocks in
that can fit in the current grid.  These blocks will always have shape (3, 3).
3.  action_mask:  numpy array (bool) of shape (num_blocks, 4, num_rows-2, num_cols-2),
representing which actions are possible given the current state of the grid.  The first
index indicates the block index, the second index indicates the rotation index, and the
third and fourth indices indicate the row and column coordinate of where a blocks top
left-most corner may be placed respectively.  These values will always be num_rows-2 and
num_cols-2 respectively to make it impossible to place a block outside the current grid.
It returns a numpy array of size (num_blocks, 4, num_rows-2, num_cols-2) representing how
valuable it is to place a block with a rotation with its top-left corner on the row,col
position in the grid.
When writing the new function, follow these guidelines:  Think Outside the Box:  Avoid
simply rewriting or rephrasing existing approaches.  Prioritize creating novel solutions
rather than making superficial tweaks.  Analyze the Score Drivers:  Analyze the
characteristics of the higher-scoring function.  Identify what it is doing differently
or more effectively than the lower-scoring function.  Determine which specific changes or
techniques lead to better performance.
To summarize, your task is to write a new function named priority() that will perform
better than both functions above and achieve a higher score.
```

Prompt 4: Description of the flatpack problem.

## A.8 Experimental details

In our experimental setup, we maintain a database of programs consisting of six islands. For prompt construction, we use $m = 2$ programs, and we generate $K = 8$ outputs for every prompt. We set the reinforcement learning frequency parameter to $f_{\mathrm{RL}} = 400$, which results in alternating between the two phases after generating 3,200 outputs. We run our experiments up to the timestep $T = 2800$, which corresponds to a total of approximately 22,400 output samples from the LLM.

**Sampling parameters** For each query to the LLM, we generate $K = 8$ outputs using a temperature of 0.9, top-$k$ sampling with $k = 100$, and nucleus sampling with $p = 0.95$ (Holtzman et al., 2019). The generated outputs are constrained to a maximum length of 2048 tokens. For inference, we utilize the Text Generation Inference (TGI) implementation (Hugging Face, 2025).

**Training Parameters** For DPO training, we apply a regularization strength of $\beta = 0.4$. Each training phase consists of 2 epochs and the AdamW (Loshchilov & Hutter, 2017) optimizer. In addition to the learning rate schedule across timesteps $t$, we use a cosine learning rate schedule inside each training phase, where the learning rate obtained by the timestep schedule is used as the starting learning rate. The learning rate is optimized for

every model on the validation set via a grid search sweep over the range $[3 \times 10^{-5}, 5 \times 10^{-7}]$. Once we found good training hyperparams for the bin packing problem, we used the same ones to perform experiments on the other two benchmarks, without further tuning. For memory-efficient fine-tuning of the model, we utilized LoRA adapters (Hu et al., 2021), configuring the rank to 64 and setting $\alpha$ to 32. Our implementation leverages the TRL library (von Werra et al., 2020) in combination with Accelerate (Gugger et al., 2022).

As there are fewer data points to train on in the early training phases and more data points in the late phases, it is crucial to balance training sufficiently in the initial stages while not overfitting in the final stages. To achieve this balance, we implement a learning rate schedule that decays the learning rate over time based on the timestep $t$: $\alpha_t = \alpha_{\text{init}} * \sqrt{1000/t}$.

**Constraints on programs** To prevent scenarios where the LLM might produce non-terminating or excessively time-consuming programs, we establish maximum execution times of 60 seconds for bin packing and flatpack tasks, and 90 seconds for the traveling salesman problem. Additionally, to avoid excessive memory consumption, we limit the memory usage to 5 GB.

### A.9 Results on Hashcode competition problems and LLM-SR benchmarks

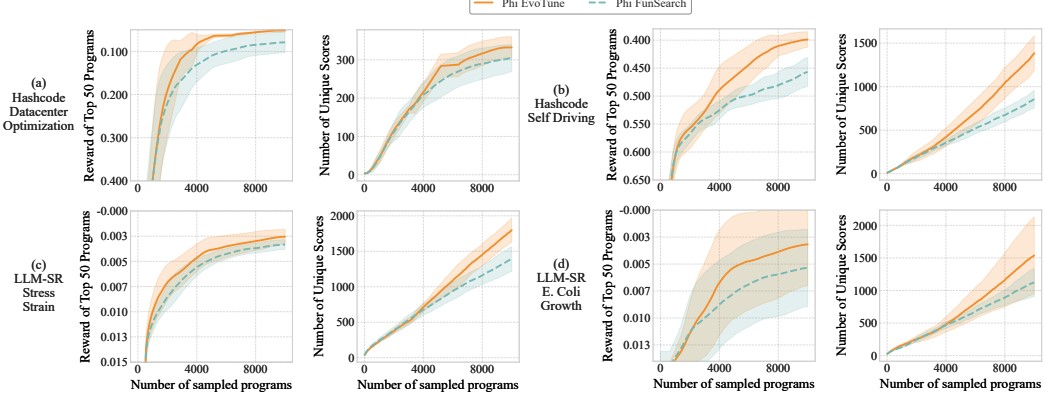

Figure 4: **Results on two Hashcode problems, and two problems from LLM-SR.** EvoTune consistently outperforms FunSearch across all problems in terms of discovering higher scoring best 50 solutions and higher diversity as measured by number of discovered solutions with unique scores. Shaded regions show standard deviation across 4 seeds.

**Hashcode results** In Figure 4, we show the performance of our method against FunSearch on two complex real-world problems from the Google HashCode competition. These tasks offer a testbed to evaluate whether our evolved heuristics can match, or even surpass, the performance of solutions developed by top human teams in competitive programming.

We compare Evotune to a standard Funsearch-style setup using the same model (Phi-3.5 Instruct). As mentioned in the main text, on the Datacenter Optimization task, both methods are able to outperform the best solutions found by human teams in the competition. However, on Self-Driving task, neither EvoTune or FunSearch do not outperform the best solution found by human teams within the sampling budget of 20,000 functions. However both methods still achieve competitive results of reaching percentiles of 87.6% and 87.1%, respectively.

**LLM-SR results** We evaluate EvoTune on two real-world symbolic regression tasks from the LLM-SR benchmark suite (Shojaee et al., 2024): E. coli Growth and Stress-Strain Prediction. Our experimental setup mirrors that of the original work, using a comparable compute budget of approximately 10,000 program samples (equivalent to the reported

| Method | E. coli Growth (ID) | E. coli Growth (OOD) | Stress-Strain (ID) | Stress-Strain (OOD) |
|---|---|---|---|---|
| EvoTune (Phi 3.5 Mini 3.8B) | 0.0082 | 0.0322 | **0.0033** | **0.0035** |
| FunSearch (Phi 3.5 Mini 3.8B) | 0.0383 | 0.0636 | 0.0037 | 0.0074 |
| LLM-SR (Mixtral 8x7B) | **0.0026** | **0.0037** | 0.0162 | 0.0946 |
| LLM-SR (GPT-3.5-turbo) | 0.0214 | 0.0264 | 0.0210 | 0.0516 |

Table 2: Results on two problems from the LLM-SR Shojaee et al. (2024) benchmark suite. We report the Normalized Mean Squared Error (NMSE) on in-distribution (ID) and out-of-distribution (OOD) test sets; lower is better. FunSearch denotes LLM-based evolutionary search in our implementation.

2,500 iterations). As shown in Figure 4, EvoTune consistently achieves better performance than our FunSearch baseline during the evolutionary search. As our implementation of the FunSearch baseline closely resembles the LLM-SR method, the performance advantage of EvoTune in both Figure 4 and Table 2 can be attributed to the addition of its reinforcement learning fine-tuning stage to a standard LLM-based evolutionary search.

Throughout the equation evolution on the training dataset, Evotune outperforms FunSearch as shown in Figure 4. As our implementation of FunSearch baseline closely parallels the LLM-SR method, the performance advantage of EvoTune over FunSearch in both Figure 4 and Table 2 can be interpreted as the benefit of adding a reinforcement learning fine-tuning stage to the LLM-SR method.

For the final evaluation, we take the best performing program found on the training set and evaluate it on in-distribution (ID) and out-of-distribution (OOD) test sets. Table 2 reports the Normalized Mean Squared Error (NMSE), where lower values are better. The results demonstrate that EvoTune is highly competitive, despite utilizing a significantly smaller model (Phi-3.5 Mini, 3.8B parameters) compared to the LLM-SR baselines (Mixtral 8x7B and GPT-3.5-turbo). Notably, on the Stress-Strain task, EvoTune outperforms both larger models on the ID and OOD test sets and remains competitive with the other methods on E.Coli Growth.

## A.10 Additional results for bin packing, travelling salesman problem, and flatpack

Beyond the results presented in Section 4.2, we provide additional insights into the performance of our method.

| | Validation Set | | | Validation-Perturbed Set | | | Test Set | | |
|---|---|---|---|---|---|---|---|---|---|
| | 9.6k | 16k | 22.4k | 9.6k | 16k | 22.4k | 9.6k | 16k | 22.4k |
| **Bin Packing** | | | | | | | | | |
| Llama FunSearch | 4.44 ± 0.18 | 3.99 ± 0.17 | 3.61 ± 0.15 | 4.13 ± 0.20 | 3.76 ± 0.19 | 3.33 ± 0.17 | 4.21 ± 0.19 | 3.79 ± 0.18 | 3.45 ± 0.15 |
| Llama EvoTune | **4.18 ± 0.19** | **3.36 ± 0.14** | **3.23 ± 0.15** | 3.98 ± 0.19 | **3.14 ± 0.17** | **3.01 ± 0.17** | **4.01 ± 0.17** | **3.25 ± 0.14** | **3.13 ± 0.15** |
| Phi FunSearch | 3.69 ± 0.18 | 3.27 ± 0.12 | 3.03 ± 0.06 | 3.40 ± 0.17 | 2.98 ± 0.12 | 2.78 ± 0.06 | 3.48 ± 0.16 | 3.07 ± 0.11 | 2.92 ± 0.05 |
| Phi EvoTune | **3.25 ± 0.14** | **3.01 ± 0.11** | **2.80 ± 0.11** | **2.95 ± 0.14** | **2.67 ± 0.11** | **2.48 ± 0.11** | **3.06 ± 0.13** | **2.81 ± 0.11** | **2.59 ± 0.13** |
| Granite FunSearch | 3.15 ± 0.06 | 3.07 ± 0.06 | 2.96 ± 0.08 | 2.91 ± 0.08 | 2.81 ± 0.06 | 2.75 ± 0.08 | 3.03 ± 0.05 | 2.93 ± 0.07 | 2.84 ± 0.09 |
| Granite EvoTune | **3.00 ± 0.07** | **2.91 ± 0.07** | **2.85 ± 0.07** | **2.75 ± 0.09** | **2.66 ± 0.08** | **2.56 ± 0.08** | **2.91 ± 0.07** | **2.80 ± 0.07** | **2.73 ± 0.08** |
| **Traveling Salesman Problem** | | | | | | | | | |
| Llama FunSearch | 2.545 ± 0.005 | 2.533 ± 0.006 | 2.525 ± 0.006 | 2.910 ± 0.003 | 2.898 ± 0.005 | 2.883 ± 0.006 | 2.556 ± 0.006 | 2.547 ± 0.006 | 2.540 ± 0.005 |
| Llama EvoTune | **2.534 ± 0.005** | **2.520 ± 0.003** | **2.504 ± 0.005** | **2.895 ± 0.007** | **2.885 ± 0.007** | **2.871 ± 0.009** | 2.558 ± 0.004 | 2.544 ± 0.005 | **2.530 ± 0.005** |
| Phi FunSearch | 2.556 ± 0.004 | 2.543 ± 0.005 | 2.535 ± 0.002 | 2.913 ± 0.004 | 2.907 ± 0.005 | 2.902 ± 0.006 | 2.567 ± 0.010 | 2.559 ± 0.008 | 2.555 ± 0.008 |
| Phi EvoTune | **2.532 ± 0.009** | **2.519 ± 0.007** | **2.509 ± 0.006** | **2.908 ± 0.004** | **2.879 ± 0.008** | **2.864 ± 0.008** | **2.557 ± 0.008** | **2.546 ± 0.008** | **2.530 ± 0.006** |
| Granite FunSearch | 2.515 ± 0.008 | 2.497 ± 0.006 | 2.488 ± 0.006 | 2.874 ± 0.007 | 2.860 ± 0.006 | 2.854 ± 0.007 | **2.519 ± 0.005** | 2.506 ± 0.005 | 2.503 ± 0.006 |
| Granite EvoTune | **2.501 ± 0.006** | **2.486 ± 0.005** | **2.476 ± 0.005** | **2.869 ± 0.007** | 2.860 ± 0.007 | 2.853 ± 0.005 | **2.525 ± 0.002** | **2.508 ± 0.004** | **2.497 ± 0.005** |
| **Flatpack** | | | | | | | | | |
| Llama FunSearch | 0.144 ± 0.006 | 0.138 ± 0.005 | 0.133 ± 0.005 | 0.140 ± 0.006 | 0.132 ± 0.007 | 0.128 ± 0.006 | 0.151 ± 0.005 | 0.145 ± 0.005 | 0.138 ± 0.005 |
| Llama EvoTune | **0.133 ± 0.003** | **0.119 ± 0.003** | **0.112 ± 0.004** | **0.126 ± 0.004** | **0.111 ± 0.003** | **0.105 ± 0.004** | **0.140 ± 0.003** | **0.125 ± 0.003** | **0.120 ± 0.004** |
| Phi FunSearch | 0.126 ± 0.005 | 0.115 ± 0.001 | 0.109 ± 0.002 | 0.115 ± 0.005 | 0.102 ± 0.001 | 0.101 ± 0.001 | 0.136 ± 0.004 | 0.123 ± 0.001 | 0.116 ± 0.002 |
| Phi EvoTune | 0.124 ± 0.006 | 0.110 ± 0.005 | **0.103 ± 0.003** | 0.114 ± 0.005 | 0.104 ± 0.005 | **0.098 ± 0.002** | 0.133 ± 0.006 | **0.117 ± 0.005** | **0.107 ± 0.003** |
| Granite FunSearch | 0.109 ± 0.001 | 0.106 ± 0.001 | 0.101 ± 0.002 | 0.100 ± 0.001 | 0.098 ± 0.000 | 0.096 ± 0.000 | 0.118 ± 0.001 | 0.114 ± 0.001 | 0.109 ± 0.002 |
| Granite EvoTune | **0.107 ± 0.001** | **0.103 ± 0.001** | 0.100 ± 0.002 | 0.101 ± 0.001 | 0.097 ± 0.002 | 0.096 ± 0.002 | **0.114 ± 0.001** | **0.111 ± 0.001** | 0.109 ± 0.002 |

Table 3: **Optimality gap achieved by the single best program found.** We report the mean and standard error across 10 seeds on validation, validation-perturbed, and test sets for three sampling budgets (9.4k, 16k and 22.4k). Similarly to results in Table 1, our method outperforms the baseline in most cases.

Table 3 reports the best score achieved by a single program. This metric is more volatile than the average of the top 50 programs and despite the variability, our method still surpasses the FunSearch baseline in most cases.

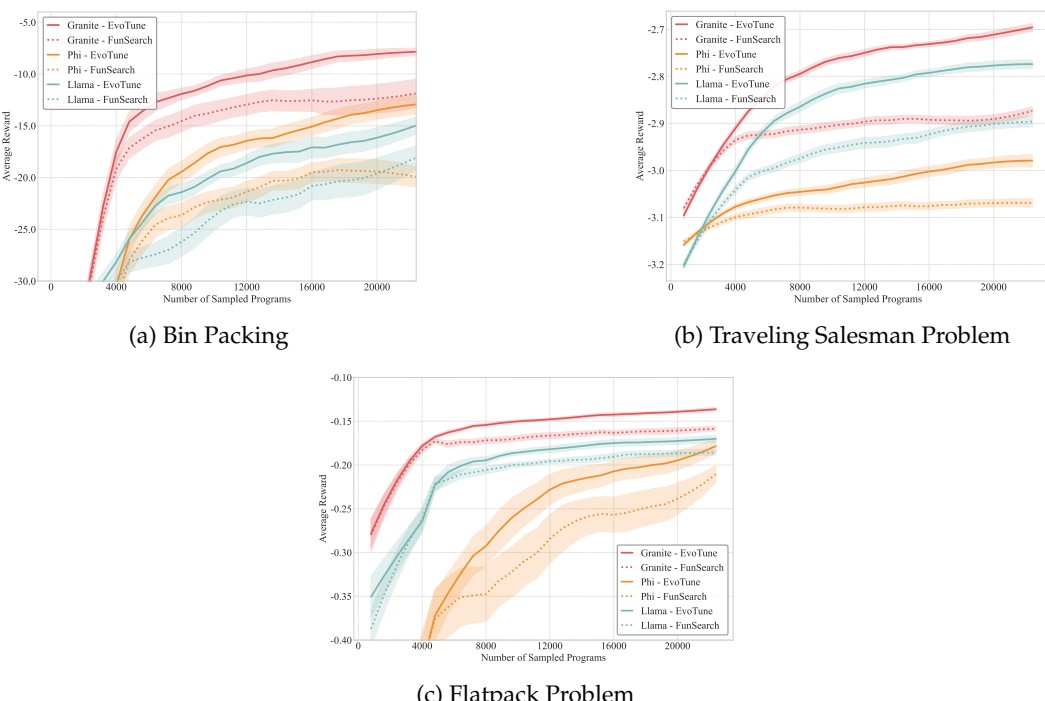

(a) Bin Packing

(b) Traveling Salesman Problem

(c) Flatpack Problem

Figure 5: **Average reward scores of valid sampled programs.** The shaded areas represent the standard error over 10 seeds. Our method outperforms the baseline in terms of its outputs having a better average score.

Figure 5 presents the average score values of the programs that pass the evaluation. The results indicate that, on average, `EvoTune` generates higher-scoring programs than the baseline, demonstrating a sustained advantage throughout the search process.

We analyze the distribution of optimality gap scores across different sampling budgets for programs generated by various models in BP (Figure 6), TSP (Figure 7) and FP (Figure 8). Initially, both `EvoTune` and FunSearch yield similar optimality gap distributions. However, as the search progresses, `EvoTune` shifts its distribution more significantly towards lower optimality gaps by discovering a greater number of high-scoring programs. As most of the increase occurs in the high-performing region, this validates the effectiveness of the RL-augmented search mechanism.

In summary, `EvoTune` effectively guides program generation toward high-scoring solutions, achieving superior performance compared to the baseline by more rapidly discovering higher scoring solutions.

In addition to performance evaluation, we analyze the structural and semantic organization of the functions discovered by both `EvoTune` and FunSearch. To this end, we embed all generated functions into a semantic embedding space using a pre-trained NeoBERT (Breton et al., 2025) encoder, and visualize the resulting representations via t-SNE (Van der Maaten & Hinton, 2008). Figure 9a and Figure 9b show t-SNE visualizations for `EvoTune` and FunSearch across three tasks. In the top rows, functions are colored by their assigned island in the program database; in the bottom rows, coloring reflects the timestep of their discovery. Both methods exhibit structured clusters that progressively diverge from the initialization function as the sampling budget increases. Notably, functions within the same island tend to display greater semantic similarity compared to those across different islands.

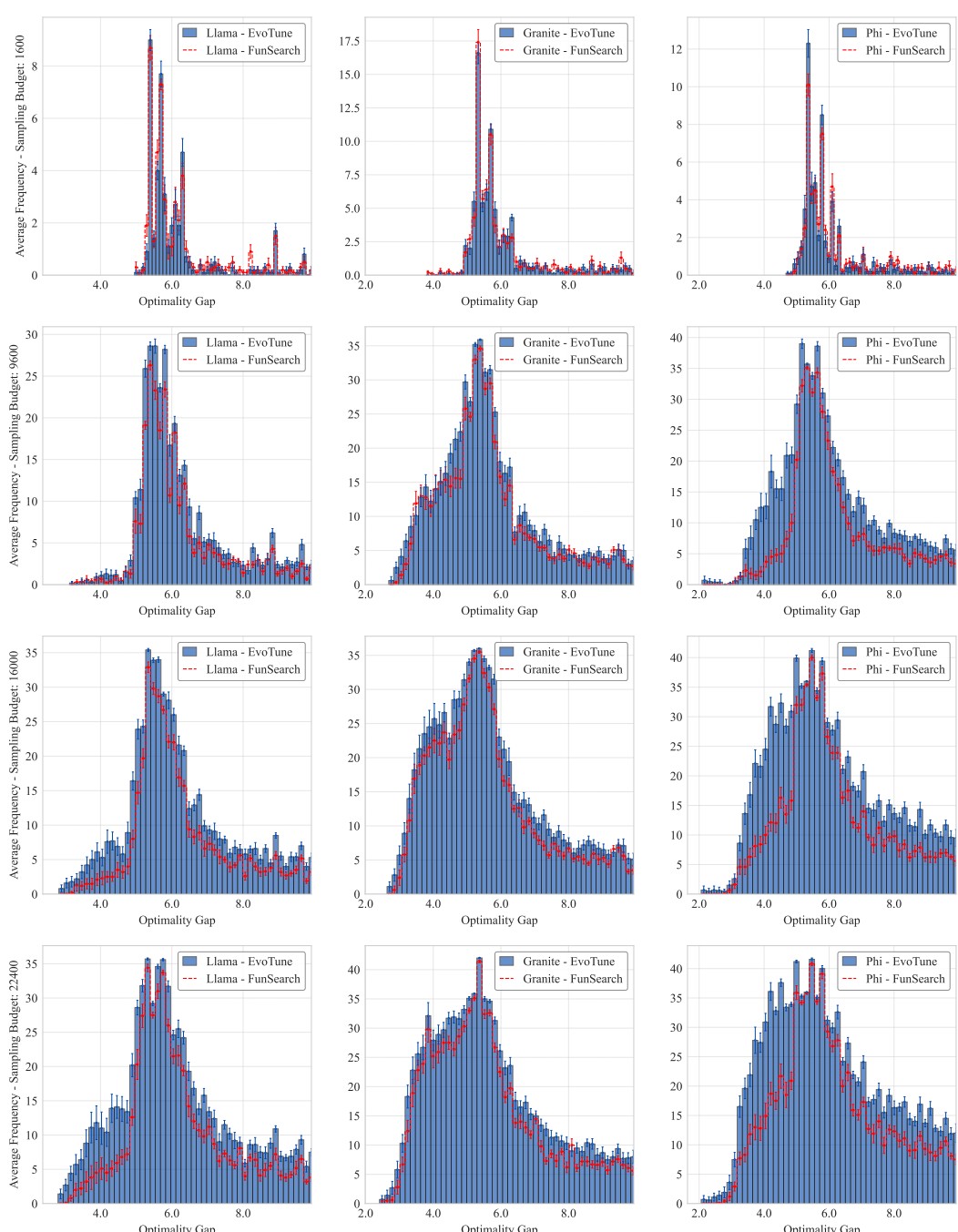

Figure 6: Histograms showing the distribution of scores in the program database on the **BP** task for four checkpoints and all three models. Results are averaged across 10 seeds. `EvoTune` outperforms FunSearch in steering the policy towards high-performing regions of the search space.

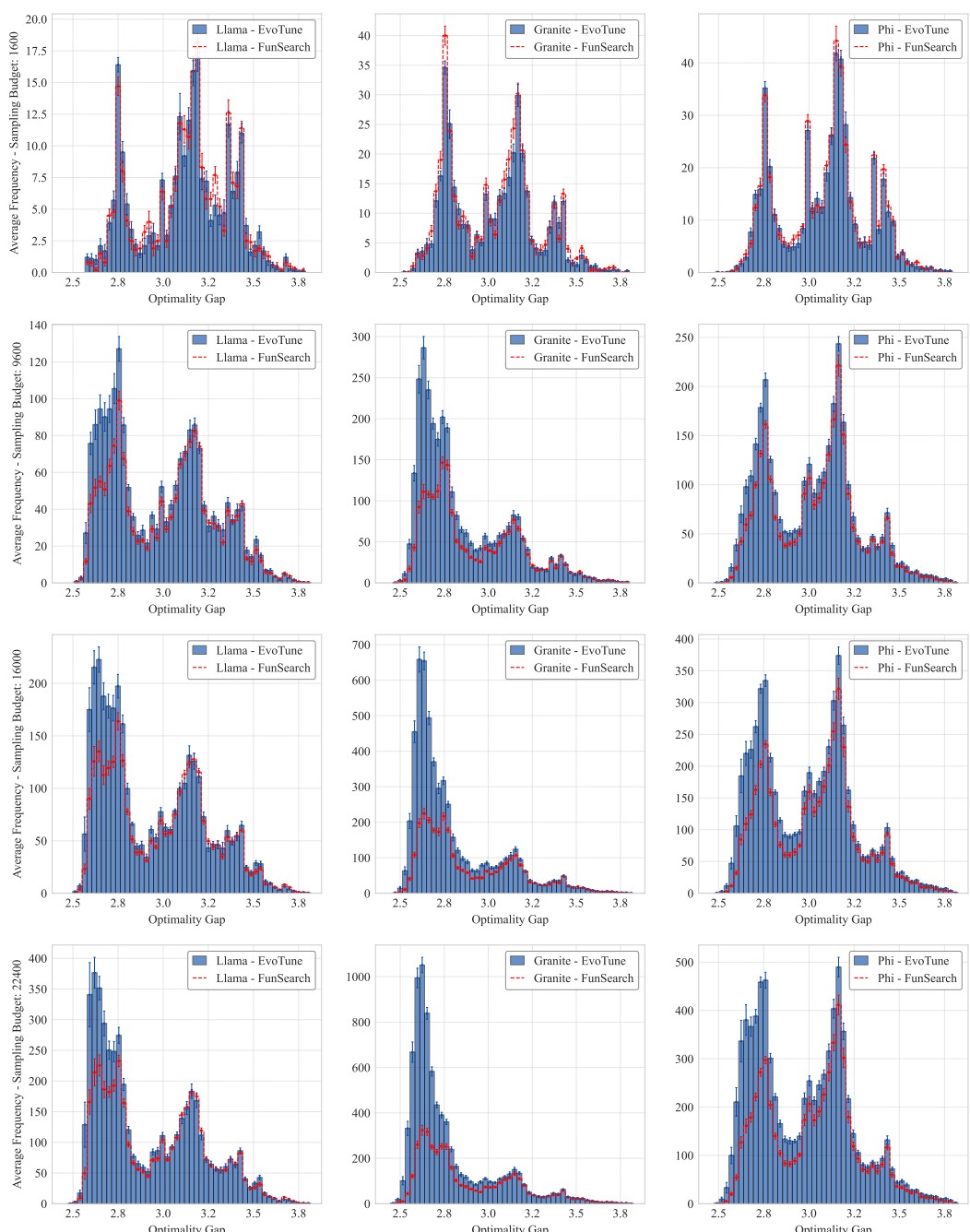

Figure 7: Histograms showing the distribution of scores in the program database on the **TSP** task for four checkpoints and all three models. Results are averaged across 10 seeds. `EvoTune` outperforms FunSearch in steering the policy towards high-performing regions of the search space.

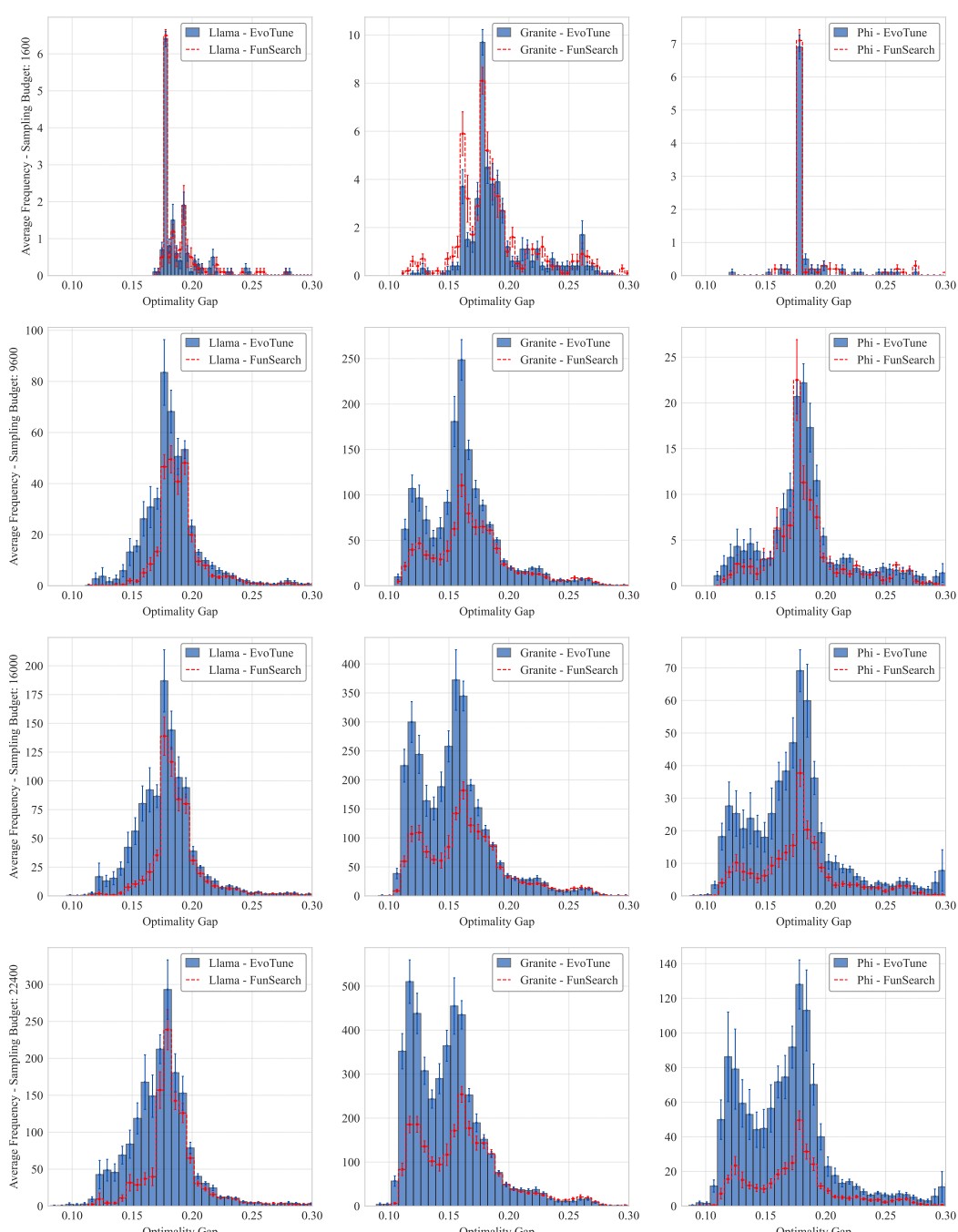

Figure 8: Histograms showing the distribution of scores in the program database on the **FP** task for four checkpoints and all three models. Results are averaged across 10 seeds. `EvoTune` outperforms FunSearch in steering the policy towards high-performing regions of the search space.

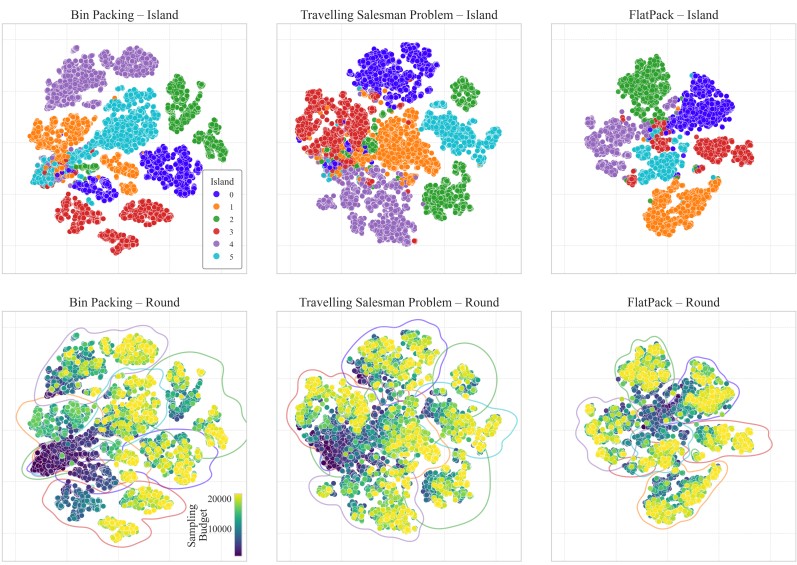

(a) t-SNE visualizations of function embeddings produced by `EvoTune` using representations from a pre-trained NeoBERT encoder. The top row is colored by program database island, while the bottom row is colored with increasing sampling budget. For each task, functions are taken from the best-performing model and seed. `EvoTune` reveals structured clusters that divert from the initialization function over increasing sampling budget.

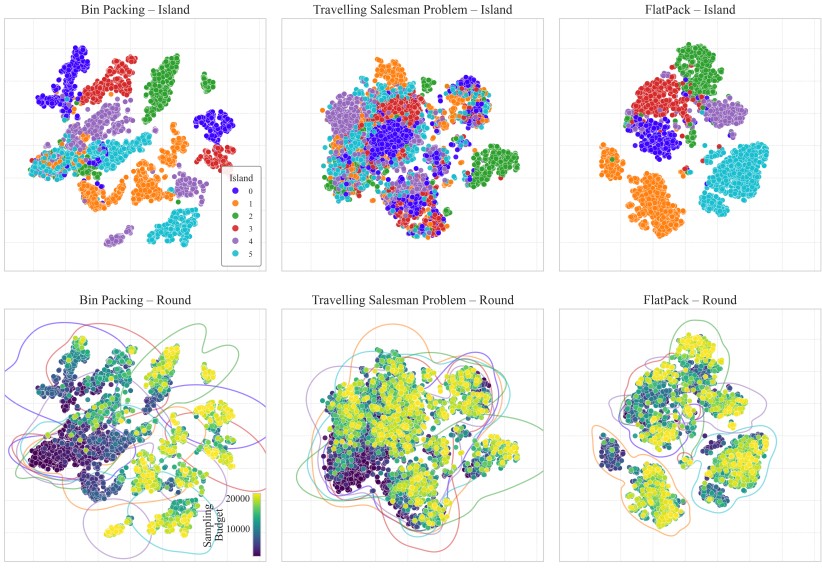

(b) t-SNE visualizations of function embeddings from FunSearch using the same setup as Figure 9a.

Figure 9: **Comparison of t-SNE visualizations for `EvoTune` (*Top*) and FunSearch (*Bottom*) across three tasks.**

## A.11 Generated programs

For completeness, we present the best heuristic found for BP, TSP, and FP in Listing 1, 2, and 3. These heuristics are found by Phi 3.5 Instruct 3.8B, Granite 3.1 2B Instruct and Llama3.2 1B Instruct, respectively.

```python
import numpy as np
def priority(item: float, bins: np.ndarray, decay_rate: float = 1.2, load_balance_weight:
↪  float = 0.5,
            balance_threshold: float = 0.05, max_balance_bonus: float = 7.0,
            ↪  urgency_inflation_rate: float = 1.3,
            innovation_factor: float = 1.5, dynamic_state_weight: float = 0.25, time_weight:
            ↪  float = 0.1,
            real_time_optimization_step: float = 0.01, history_decay_rate: float = 0.95,
            urgency_trend_weight: float = 0.2, bin_state_adaptation_rate: float = 0.05,
            capacity_sensitivity_factor: float = 1.1, exploration_factor: float = 0.05,
            ↪  exploration_decay: float = 0.99,
            temporal_diversity_weight: float = 0.07) -> np.ndarray:
    """
    An innovative priority calculation function that not only builds upon the advanced
    ↪  strategies of the previous versions but
    also incorporates real-time adaptive learning and forecasting to anticipate future bin
    ↪  states, ensuring optimal bin allocation.

    Args:
    item: Size of the item to be added to the bin.
    bins: Array of capacities for each bin.
    decay_rate: Rate of exponential decay; higher values increase sensitivity to capacity
    ↪  differences.
    load_balance_weight: Influence of load balancing on the priority score.
    balance_threshold: Threshold below which the bin balance is considered insufficiently
    ↪  balanced.
    max_balance_bonus: Maximum bonus for a perfectly balanced bin.
    urgency_inflation_rate: Rate at which urgency increases for bins closer to capacity.
    innovation_factor: Multiplier for balance, urgency impact, and dynamic state.
    dynamic_state_weight: Weight given to the dynamic bin state, such as historical
    ↪  performance.
    time_weight: Weight for incorporating the time factor into optimization.
    real_time_optimization_step: Adjustment factor in real-time optimization.
    history_decay_rate: Decay factor for reducing the weight of historical performance over
    ↪  time.
    urgency_trend_weight: Weight to emphasize urgency trends in the bin allocation strategy.
    bin_state_adaptation_rate: Rate at which the bin state adaptation influences priority
    ↪  scores.
    capacity_sensitivity_factor: Multiplier that amplifies the effect of bin capacity on
    ↪  priority scores.
    exploration_factor: Weight given to unused bin space as a priority.
    exploration_decay: Decay factor for reducing the influence of exploration over time.
    temporal_diversity_weight: Weight given to diversity in usage across time for
    ↪  optimization.

    Returns:
    Array of priority scores for each bin aiming for optimal strategic allocation.
    """

    # Calculate ideal capacity and balance factor
    ideal_capacity = np.mean(bins)
    balance_factor = np.where(np.abs(bins - ideal_capacity) <= balance_threshold, 1, (1 / (1 +
    ↪  np.abs(bins - ideal_capacity) / balance_threshold)))

    # Calculate urgency bonus
    urgency_bonus = np.where(bins - item >= balance_threshold, urgency_inflation_rate, 1)

    # Apply time influence for real-time optimization
    time_influence = np.sin(np.arange(len(bins)) * real_time_optimization_step)

    # Calculate adaptive decay considering urgency and time influence
    adaptive_decay = -(np.abs(bins - item) * decay_rate ** (np.abs(bins - item) *
    ↪  urgency_bonus * time_influence)) * balance_factor

    # Calculate load balance score and exploration bonus
    load_balance_score = np.clip(np.std(bins) / np.mean(bins) * load_balance_weight, 0, 1)
    exploration_bonus = np.clip(1 - np.exp(-np.sum(bins - item) / np.sum(bins) *
    ↪  exploration_factor), 0, 1)

    # Introduce a capacity sensitivity factor
    capacity_sensitivity = np.power(np.max(bins) / np.min(bins), capacity_sensitivity_factor)
```

```python
    # Calculate temporal diversity
    temporal_diversity = np.exp(-np.arange(len(bins)) / np.max(bins) *
    ↪   temporal_diversity_weight)

    # Calculate dynamic state impact
    bin_state_impact = np.exp(-np.var(bins) * dynamic_state_weight) * temporal_diversity

    # Combine all factors with emphasis on dynamic states, urgency sensitivity, load
    ↪   balancing, exploration, and capacity sensitivity
    priority_scores = adaptive_decay * capacity_sensitivity
    priority_scores += load_balance_score * max_balance_bonus
    priority_scores += bin_state_impact
    priority_scores += exploration_bonus * exploration_factor

    # Normalize and scale scores for real-time optimization, considering historical
    ↪   performance decay, urgency, temporal diversity, and capacity sensitivity
    priority_scores = np.clip(priority_scores, 0, 1) * (1 + np.log1p(np.sum(bins - item))) *
    ↪   innovation_factor * time_weight

    return priority_scores
```

Listing 1: The highest scoring program discovered for *bin packing problem*. This program was generated by `EvoTune` with Phi 3.5 Instruct model and it achieves an optimality-gap of 2.06.

```python
import numpy as np
def heuristics(distance_matrix):
    num_nodes = distance_matrix.shape[0]

    # Average Distance and Connectivity
    avg_distances = np.mean(distance_matrix, axis=1)
    local_connectivity = np.sum(distance_matrix, axis=1) / (num_nodes - 1)
    global_connectivity = np.sum(distance_matrix) / (num_nodes * (num_nodes - 1))

    # Adaptive Shortcut Factor
    adaptive_shortcut_factor = np.maximum(avg_distances, 0.5) / np.max(avg_distances)

    # Hierarchical Complexity
    hierarchical_complexity = np.sum(distance_matrix ** 2, axis=1)

    # Node Importance Factor
    node_importance = np.sum(distance_matrix, axis=1)

    # Dynamic Influence
    influence_factor = 1 / (1 + np.exp(-distance_matrix / 10))  # Gaussian decay

    # Local and Global Connectivity Adjustment
    local_density = 1 / np.sum(distance_matrix ** 2, axis=1)
    local_connectivity_factor = np.minimum(1, np.exp(-local_density))  # Adjusted for local
    ↪   node importance

    # Novel Dynamic Decay: Adaptive Local Density Adjustment
    # This factor gives more weight to less densely connected nodes
    local_density_factor = np.minimum(1, 1 / local_connectivity)

    # Popularity Factor
    popularity_factor = np.sum(np.power(distance_matrix, 2), axis=1) / np.sum(distance_matrix,
    ↪   axis=1)

    # Novel Factor: Edge-wise Connectivity
    edge_connectivity = np.copy(distance_matrix)
    for k in range(num_nodes):
        edge_connectivity[k] = np.sum(distance_matrix[k]) / (num_nodes - 1)

    # High-Degree Weight
    high_degree_weight = 0.5  # Adjusted to emphasize high-degree nodes
    heuristic_matrix = (distance_matrix ** 2) * (1 - avg_distances) * (1 -
    ↪   adaptive_shortcut_factor) \
                        * (1 - hierarchical_complexity) * (1 - node_importance) *
                        ↪   high_degree_weight \
                        * np.maximum(avg_distances, 0.5)  # Favor high-degree nodes

    # Time Stability Factor
    time_stability = np.exp(-distance_matrix / 100)  # Adjusted for edges with larger time
    ↪   differences
    heuristic_matrix *= time_stability
```

```python
    # Novel Factor: Edge-wise Connectivity
    # This factor considers local centrality diversity and edge-wise connectivity
    edge_diversity = np.abs(np.minimum(local_connectivity, edge_connectivity) \
                  - np.maximum(local_connectivity, edge_connectivity))
    edge_connectivity_factor = 1 - edge_diversity

    # Combine all factors
    heuristic_matrix *= (1 - local_density_factor) - influence_factor - popularity_factor \
                      - edge_connectivity_factor - time_stability

    # Normalization to ensure the heuristic matrix values sum to 1 for each row (each edge)
    heuristic_matrix /= np.sum(heuristic_matrix, axis=1)[:, np.newaxis]

    # Add a novel factor: Temporal Stability Factor
    temporal_stability = np.exp(-distance_matrix / 1000)  # Adjusted for edges with older time
    ↪ differences
    heuristic_matrix *= temporal_stability

    return heuristic_matrix
```

Listing 2: The highest scoring program discovered for *traveling salesman problem*. This program was generated by `EvoTune` with Granite 3.1 2B Instruct model and it achieves an optimality gap of 2.446.

```python
import numpy as np
import numpy.lib.stride_tricks as st
import math
from typing import Tuple, Union
def priority(
        current_grid: np.ndarray,
        blocks: np.ndarray,
        action_mask: np.ndarray
) -> np.ndarray:
    # Precompute rotated versions of all blocks
    num_blocks = blocks.shape[0]
    rotated_blocks = np.array([
        [np.rot90(block, k=r) for r in range(4)] for block in blocks
    ])

    # Pad the grid once (for boundary checking)
    padded_grid = np.pad(current_grid, 1, mode='constant', constant_values=0)

    # Initialize Q-value matrix
    values = np.full(action_mask.shape, -np.inf, dtype=np.float32)

    # Compute scores for each possible placement of a block with a rotation that has been
    ↪ blocked
    for block_idx in range(num_blocks):
        for rotation in range(4):
            block = rotated_blocks[block_idx, rotation - 1]  # Subtract 1 to adjust rotation
            ↪ index
            block_rows, block_cols = block.shape

            # Extract all possible placements using NumPy slicing
            sub_grids = np.lib.stride_tricks.sliding_window_view(padded_grid, (block_rows - 1,
            ↪ block_cols - 1))

            # Compute the score for each placement
            scores = []
            for i in range(block_rows - 1):
                for j in range(block_cols - 1):
                    # Extract top-left corner of the block
                    if block_idx == block_idx:
                        top_left = block[i:i+2, j:j+2]
                    else:
                        top_left = None
                    score = np.sum(np.where(top_left, 1, 0)) * (block_rows - 1) * (block_cols
                    ↪ - 1)
                    scores.append(score)

            # Compute the weighted sum of the scores for blocks with a rotation that has been
            ↪ blocked
            weights = np.sqrt(block_rows * block_cols) / (2 ** (block_rows - 1) * (2 **
            ↪ (block_cols - 1)))
```

```
        weighted_sum = np.sum([weights * scores for scores in scores])
        values[block_idx, rotation - 1, ...] = weighted_sum

    # Apply action mask in one operation
    values[~action_mask] = -np.inf

    # Calculate absolute values of Q-Values for all blocks
    abs_values = np.abs(values)

    # Calculate cumulative sum
    cum_sum = np.cumsum(abs_values, axis=2)

    return cum_sum
```

Listing 3: The highest scoring program discovered for the *flatpack* problem. This program was generated by `EvoTune` with the Llama3.2 1B Instruct model and it achieves an optimality gap of 0.0829.

### A.12    Comparison to non-LLM-based methods

The primary goal of our work was not to achieve state-of-the-art performance on specific benchmarks, especially as we use smaller, open-source LLMs, but rather to use these tasks as testbeds for rigorously evaluating the effectiveness of our proposed EvoTune method compared to the baseline with no training. Nevertheless, to contextualize its performance, we benchmark EvoTune against specialized non-LLM methods. For the TSP, we compare it against methods specifically designed for this problem, and for bin packing and flatpack problems, we use human-designed heuristics as baselines.

**Comparison to task-specific methods (traveling salesman problem)**    We compare Evo-Tune with the following methods specialized for the traveling salesman problem:

- LEHD (Luo et al., 2023) is a neural solver successor to POMO (Kwon et al., 2020) and Attention Model (Kool et al., 2018), which augments supervised learning of heuristics with a decoder specialized for TSP

- KGLS (Arnold & Sörensen, 2019) is a search-based baseline - a more advanced version of the Guided Local Search (GLS).

- NeuralGLS (Sui et al., 2024) is a hybrid model integrating GLS with graph convolutional networks specialized for TSP

We evaluate EvoTune on 29 TSP instances from TSPLib, comprising real-world TSP instances of varying difficulty. In this way, we can directly compare with the results reported in (Luo et al., 2023; Arnold & Sörensen, 2019; Sui et al., 2024). Additionally, this evaluation serves as an additional benchmark for EvoTune to test the robustness and generalization of the evolved heuristics, since no instances from TSPLib are shown to EvoTune during training. Evaluation was done on the top-performing program evolved by Granite 3.1 as the base model.

For TSP, EvoTune evolves the heuristic that is used by guided local search (GLS), and GLS operation can be budgeted to call local search $t_{max}$ times. While the specialized baselines operate with a large budget of $t_{max} = 1000$, EvoTune was trained with a minimal budget of just $t_{max} = 20$ to reduce evaluation times due to computational constraints. In the table below, we present the performance of EvoTune's heuristic when deployed with $t_{max} \in 100, 200, 1000$, i.e., it was never optimized to find heuristics under such budgets.

The results, presented in Table 4, confirm the strong generalization capabilities of the evolved heuristic. Evotune successfully achieves near-optimal performance when it is provided the same (or even less) local search budget as the baselines. It already performs comparably on average to other baselines at $t_{max} \in 100, 200$, and with $t_{max} = 1000$ it achieves the best average across all TSPLib instances, outperforming all other methods.

**Comparison to human-designed heuristics (bin packing and flatpack problem**    For bin packing and flatpack problems, we include direct comparison to human-designed heuristics. For bin packing, we report results for the well-established best-fit heuristic. For flatpack, we use a greedy heuristic that promotes compact, efficient placements by placing larger

| Instance | POMO | LEHD | NeuralGLS | KGLS | EvoTune $t_{max} = 100$ | EvoTune $t_{max} = 200$ | EvoTune $t_{max} = 1000$ |
|---|---|---|---|---|---|---|---|
| eil51 | 0.83 | 1.64 | 0.00 | 0.67 | 0.70 | 0.70 | 0.67 |
| berlin52 | 0.04 | 0.03 | 0.00 | 0.03 | 0.03 | 0.03 | 0.03 |
| st70 | 0.31 | 0.33 | 0.00 | 0.31 | 0.31 | 0.31 | 0.31 |
| eil76 | 1.18 | 2.54 | 0.00 | 1.18 | 1.64 | 1.18 | 1.18 |
| pr76 | 0.00 | 0.22 | 0.82 | 0.00 | 0.00 | 0.00 | 0.00 |
| rat99 | 2.39 | 1.10 | 0.72 | 0.68 | 1.24 | 1.24 | 0.68 |
| kroA100 | 0.41 | 0.12 | 0.03 | 0.06 | 0.02 | 0.02 | 0.02 |
| kroB100 | 0.32 | 0.26 | 0.88 | 0.25 | 0.25 | 0.25 | 0.25 |
| kroC100 | 0.18 | 0.32 | 1.77 | 0.01 | 1.25 | 0.83 | 0.01 |
| kroD100 | 0.84 | 0.38 | 0.00 | 0.00 | 0.30 | 0.07 | 0.00 |
| kroE100 | 0.45 | 0.43 | 1.05 | 0.07 | 0.49 | 0.17 | 0.17 |
| rd100 | 0.01 | 0.01 | 0.00 | 0.02 | 0.01 | 0.01 | 0.01 |
| eil101 | 1.84 | 2.31 | 0.36 | 2.07 | 2.07 | 2.07 | 1.78 |
| lin105 | 0.52 | 0.34 | 0.65 | 0.03 | 0.03 | 0.03 | 0.03 |
| pr107 | 0.52 | 11.24 | 0.81 | 0.00 | 0.40 | 0.00 | 0.00 |
| pr124 | 0.60 | 1.11 | 0.08 | 0.08 | 0.08 | 0.08 | 0.00 |
| bier127 | 13.72 | 4.76 | 2.73 | 0.42 | 0.57 | 0.57 | 0.57 |
| ch130 | 0.16 | 0.55 | 1.19 | 0.01 | 0.01 | 0.01 | 0.01 |
| pr136 | 0.93 | 0.45 | 2.32 | 0.24 | 2.88 | 2.88 | 0.81 |
| pr144 | 0.53 | 0.19 | 0.74 | 0.00 | 0.38 | 0.09 | 0.00 |
| ch150 | 0.53 | 0.52 | 2.49 | 0.04 | 0.68 | 0.44 | 0.32 |
| kroA150 | 0.70 | 1.40 | 0.77 | 0.17 | 2.54 | 2.54 | 0.65 |
| kroB150 | 1.17 | 0.76 | 3.11 | 0.08 | 1.04 | 1.04 | 0.04 |
| pr152 | 1.05 | 12.14 | 0.00 | 0.19 | 1.03 | 0.64 | 0.00 |
| u159 | 0.95 | 1.13 | 0.90 | 0.96 | 1.44 | 1.44 | 0.00 |
| rat195 | 8.15 | 1.42 | 0.48 | 0.97 | 0.91 | 0.91 | 0.66 |
| d198 | 17.29 | 9.23 | 1.28 | 0.31 | 0.90 | 0.90 | 0.71 |
| kroA200 | 1.58 | 0.64 | 0.86 | 0.71 | 0.56 | 0.23 | 0.20 |
| kroB200 | 1.44 | 0.16 | 3.74 | 0.89 | 1.99 | 1.43 | 0.16 |
| Average | 2.02 | 1.92 | 0.96 | 0.36 | 0.82 | 0.69 | 0.32 |

Table 4: Performance comparison on TSPLib instances, showing the optimality gap (lower is better). The EvoTune heuristic was evolved using a minimal training budget $t_{max} = 20$. When deployed with the same budget as the specialized baselines $t_{max} = 1000$, it achieves the best average performance.

| | Human-designed heuristic | EvoTune | FunSearch |
|---|---|---|---|
| Bin Packing | 5.37 | 2.06 | 2.96 |
| Flatpack | 0.1092 | 0.0829 | 0.0898 |

Table 5: Comparison of optimality gaps against human-designed heuristics (lower is better).

blocks first and maximizing adjacency. Note that these human-designed heuristics are used to initialize the search process. As shown in Table 5, using a small open-source LLMs and a budget of 22.4k samples, both methods successfully discovered algorithms that outperform the human-designed starting points, with EvoTune consistently finding the best-scoring solutions.

### A.13 Alternative RL algorithm

In addition to our DPO-based `RL-Update`, we experimented with an alternative RL method based on the ReST$^{EM}$ algorithm (Gulcehre et al., 2023; Singh et al., 2023). This approach iteratively refines the base model through supervised fine-tuning on high-scoring outputs gathered during evolutionary search. Unlike DPO, which uses a ranking-based objective, ReST$^{EM}$ progressively filters the program database to focus on increasingly better-scoring samples and then fine-tunes the model on this refined dataset. At each RL-Update step $t$ (i.e. when $t \bmod f_{RL} = 0$), we proceed as shown in Algorithm 2. In our experiments, we set $p = 60$ and $L = 3$. The rest of the training parameters are similar to the ones described in Appendix A.8, including the learning rate schedule.

Our results in Figure 10 demonstrate that incorporating offline RL training - using either ReST$^{EM}$ or DPO - yields better performance than no training at all. The DPO-based update consistently outperforms the ReST$^{EM}$ variant across all tested learning rates, indicating that its ranking-based signal more effectively guides the model toward high-scoring programs.

---

**Algorithm 2** RL-Update using ReST$^{\text{EM}}$ algorithm

---

**Input:** Program database $\mathcal{D}^t$, base model $\pi_\theta^0$.

Set the initial threshold $\tau^{t,0}$ to the $p$-th percentile of all rewards $r(y)$ obtained from outputs generated since the previous RL-Update phase (i.e., from step $t - f_{\text{RL}}$ onward).

Let $r_{\max}^t = \max\{r(y) : (x, y, r(y)) \in \mathcal{D}^t\}$.

**for** $l = 0$ **to** $L - 1$ **do**

Construct the SFT dataset:

$$\mathcal{D}_{\text{SFT}}^t = \{(x, y) \in \mathcal{D}^t : r(y) \geq \tau^{t,l}\}.$$

Update $\theta$ by minimizing the negative log-likelihood loss:

$$\mathcal{L}(\theta) = -\mathbb{E}_{(x,y) \sim \mathcal{D}_{\text{SFT}}^t} \log \pi_\theta(y \mid x).$$

Update the threshold:

$$\tau^{t,l+1} \leftarrow \tau^{t,l} + \frac{r_{\max}^t - \tau^{t,0}}{L}.$$

**end for**

**Output:** Updated model $\pi_\theta^t$ with parameters $\theta$.

---

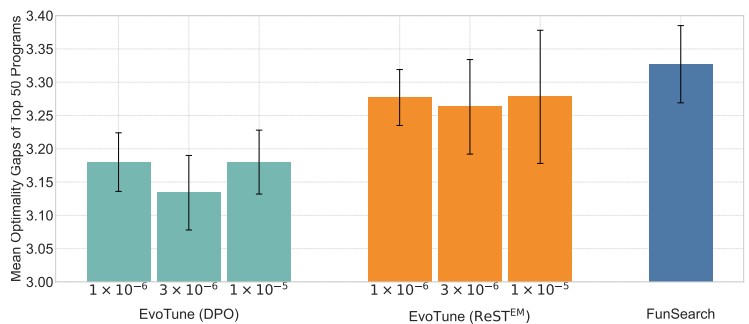

Figure 10: Mean optimality gap (lower is better) for the top 50 programs on the validation set at the final sampling budget, averaged over 10 seeds. Experiments were done using the Granite model on the bin packing problem. We compare `EvoTune` to two RL update methods - DPO and ReST$^{\text{EM}}$ - across three learning rates ($1 \times 10^{-6}$, $3 \times 10^{-6}$, and $1 \times 10^{-5}$). While both `EvoTune` variants outperform the baseline, the DPO variant achieves lower optimality gaps compared to the ReST$^{\text{EM}}$ variant. Note that for the Granite model, the learning rate of $1 \times 10^{-5}$ was used to perform DPO experiments presented in the rest of the paper.

Our early experiments also revealed that tuning ReST$^{\text{EM}}$ is challenging and its performance rapidly degrades with suboptimal hyperparameter choices. Although we focused primarily on the learning rate, which we identified as one of the most impactful hyperparameters, a more comprehensive hyperparameter sweep is needed to fully characterize the differences. Moreover, while an initial SFT phase is typically applied before DPO updates, we omitted it to reduce training time. Future work may explore hybrid approaches that combine SFT with DPO updates to further enhance performance.

