# OpenReview forum: "Algorithm Discovery With LLMs: Evolutionary Search Meets Reinforcement Learning"
_colmweb.org/COLM/2025/Conference — COLM 2025_

### Official Review · Reviewer_YtJU · 2025-04-12

**Rating:** 8
**Confidence:** 4
**Ethics Flag:** 1

**Summary:**

This paper studies LLM-based algorithm discovery for combinatorial optimization (CO). Specifically, the authors propose EvoTune, a novel approach that combines evolutionary search with RL fine-tuning (i.e., DPO) to enhance LLM-driven algorithm discovery. Overall, the paper is solid and makes a meaningful contribution to the field.

**Questions To Authors:**

* Does the `Generate K outputs [...]` step in Alg. 1 involve evolutionary search? Based on the prompt provided in Appendix A.7, it appears that no crossover or mutation operations are used.
* How is the validation dataset constructed to ensure that the optimization process remains unbiased (e.g., not overfitting to a specific data distribution)? Can the proposed method generalize to test sets containing larger-scale problem instances or those with significantly different data distributions?

**Reasons To Accept:**

* The topic studied is interesting and well-motivated.
* The paper is well-written and easy to understand.
* The proposed method is sound.

**Reasons To Reject:**

* The paper evaluates the proposed approach on only three CO tasks. Including a broader set of CO or non-CO tasks would strengthen the evaluation.
* Missing baselines. For example, comparisons with traditional TSP solvers (e.g., Concorde, LKH-3) and recently proposed neural solvers should be included.
* Evaluation on large-scale instances, such as TSP1000, is necessary to assess scalability.
* The RL fine-tuning introduces additional computational overhead. What hardware was used for the experiments? In addition to reporting the optimality gap, please also report runtime. Could you further benchmark all methods under the same time budget?

---

> ### Author Response · Authors · 2025-06-01
> **Response [1/2]**
>
> We thank the reviewer for their careful reading and positive evaluation of our work. We appreciate the recognition of the motivation, clarity, and soundness of our proposed EvoTune framework. We are glad that the overall contribution of the paper is seen as meaningful for the field.
>
> Below, we address the main concerns and questions raised:
>
> #### **1. Evaluating the method on a broader set of tasks**
>
> > *The paper evaluates the proposed approach on only three CO tasks. Including a broader set of CO or non-CO tasks would strengthen the evaluation.*
> >
>
> We appreciate the suggestion to broaden the scope of evaluation. In response, we have added results on a real-world combinatorial optimization benchmark and the non-CO task of symbolic regression (LLM-SR). These experiments further demonstrate the effectiveness and generality of our approach across diverse problem domains. We kindly refer the reviewer to the supplementary pdf and our response to all reviewers for full details and results.
>
> #### **2. Comparison to traditional solvers**
>
> > *Missing baselines. For example, comparisons with traditional TSP solvers (e.g., Concorde, LKH-3) and recently proposed neural solvers should be included.*
> >
>
> We refer the reviewer to our response to all reviewers, where we discuss this point in detail and provide supporting results.
>
> To address the reviewer’s remark on comparing with Concorde and LKH, we note that LKH is indeed the backbone of our evaluation for providing the reward signal, i.e., the optimality gaps that are provided as rewards are $\frac{C_{Evotune}-C_{LKH}}{C_{LKH}}$, where $C$ denotes the cost of the best tour returned by each method.
>
> #### **3. Evaluation on large-scale instances**
>
> > *Evaluation on large-scale instances, such as TSP1000, is necessary to assess scalability.*
> >
>
> To address the reviewer’s concern regarding the scalability of the evolved heuristics, we present the optimality gap results obtained by running the top program evolved by EvoTune and FunSearch (both with Granite as the base model) under increasing local search (LS) iteration budgets. The results are averaged across 10 instances of TSP1000 where town coordinates are uniformly sampled from $[0,1]^2$.
>
> We can observe that the heuristics obtained by evolutionary search generalize to instances of size 1000, i.e., instances that had not been seen during evolution. Moreover, the optimality gaps are comparable to those found in smaller problems (Table of the original manuscript), and are very close to the solver’s. Additionally, we note that solving these larger instances using LKH and EvoTune/FunSearch heuristics takes approximately the same time to complete (1m50s), therefore, the heuristics evolved by evolutionary search with LLMs over the course of a few hours of runtime generalize, scale, and perform comparably to the optimized solvers. This highlights the effectiveness and versatility of EvoTune when applied to different problems.
>
> | LS Budget | EvoTune | FunSearch |
> | --- | --- | --- |
> | 10 | 3.99 | 3.96 |
> | 100 | 2.92 | 2.95 |
> | 1000 | 1.48 | 1.48 |
>
> #### **4. Computational overhead of RL phase**
>
> > *The RL fine-tuning introduces additional computational overhead. What hardware was used for the experiments? In addition to reporting the optimality gap, please also report runtime.*
> >
>
> As requested by the reviewer, we include the runtime comparisons in the supplementary pdf and we kindly refer the reviewer to our common reply, where we discuss this analysis in detail. We ran our experiments on H100 and A100 GPUs.
>
> #### **5. Clarification on the “generate K outputs” step**
>
> > *Does the `Generate K outputs [...]` step in Alg. 1 involve evolutionary search? Based on the prompt provided in Appendix A.7, it appears that no crossover or mutation operations are used.*
> >
>
> Our approach does not explicitly require the use of a particular evolutionary operator. Instead, we prompt the LLM with two well-performing algorithms and instruct it to generate a new one. This leaves the nature of the transformation open-ended,  allowing the model to produce mutations, recombinations, or hybrid modifications as appropriate. We will update our manuscript to clarify this aspect of our methodology.

---

> > ### Author Response · Authors · 2025-06-01
> > **Response [2/2]**
> >
> > #### **6. Validation dataset construction and generalization**
> >
> > > *How is the validation dataset constructed to ensure that the optimization process remains unbiased (e.g., not overfitting to a specific data distribution)?*
> > >
> >
> > In addition to reporting results on a held-out test set, we also include a perturbed validation set designed to simulate domain shifts. Details on dataset construction are provided in Appendix A.1. In a nutshell:
> >
> > - **Test Set:** For each problem, we sample new instances from the same distribution as the validation set, evaluating on unseen but in-distribution data.
> > - **Perturbed Set:** To further assess generalization, we introduce domain shifts in the validation set:
> >     - TSP: modify the adjacency matrix by assigning prohibitively high costs to selected edges, making certain connections untraversable (simulating road closures or network disruptions).
> >     - Bin Packing: randomly perturb the order of items to be packed.
> >     - FlatPack: introduce an obstacle at the center of the grid where no blocks can be placed.
> >
> > Please also see our response to all reviewers for additional generalization and robustness results for TSP on TSPLib, a collection of real-world TSP instances that is not presented to EvoTune during training.
> >
> > Our intention with these additional evaluation datasets was to explore how much the improvements gained by EvoTune in the original search space would translate to related problem instances. **We were encouraged to observe that the gains generalized to these similar search spaces.**
> >
> > > *Can the proposed method generalize to test sets containing larger-scale problem instances or those with significantly different data distributions?*
> > >
> >
> > To assess OOD generalization to larger problem instances, we benchmarked the performance on the TSP dataset with 1000 nodes, which we detailed earlier in this response. For researchers interested in achieving even stronger OOD generalization, we suggest incorporating many different problem sizes within the validation dataset (for TSP, we currently only include two - 100 nodes and 200 nodes graphs). This already disincentivizes the discovered algorithms from becoming specialized to particular problem sizes and has encouraged solutions that generalize more broadly across scales, as can be observed by our results on TSP1000 and various TSPLib instances.
> >
> > The primary focus of our work is on improving search efficiency and solution quality within a fixed search regime, rather than maximizing OOD generalization. In some problems, mathematicians are only concerned with optimizing the solution for a specific configuration or dimension. However, the additional experiments we carried out, as well as those suggested by the reviewer, help showcase the generalizability of evolved heuristics as a by-product of our method.

---

> > > ### Comment · Reviewer_YtJU · 2025-06-03
> > >
> > > Thank you for your response, which adequately addresses my comments. I will finalize my evaluation as a `clear accept`. One final note: as this field is rapidly evolving, I recommend that the authors discuss more recent works, such as those summarized in [this repository](https://github.com/ai4co/awesome-fm4co), in the final version.

---

> > > > ### Author Response · Authors · 2025-06-04
> > > >
> > > > Thank you once again for your thoughtful feedback. We are pleased that our response addressed your concerns and are grateful for the improved evaluation. We will ensure that the final version includes a discussion of the most recent works referenced in the repository you mentioned.

---

### Official Review · Reviewer_rxQm · 2025-05-13

**Rating:** 7
**Confidence:** 4
**Ethics Flag:** 1

**Summary:**

- Quality: The algorithm is clearly specified and includes sufficient testing scenario, while it is still doubtful if the comparison is complete and makes sense.
- Clarity: The paper has very high clarity. It explains the algorithm and everything very well.
-  Originality: This is not the first work combining LLM with evolutionary search for algorithm discovery. The only novelty is maybe using RL. (see Reasons To Reject)
- Significance:  I don't think the paper solves broader tasks. The paper is evaluated only on three well-known combinatorial optimization tasks without any extension to other meaningful tasks. For example, LLM-SR is more on the realistic scenario than this paper.

**Questions To Authors:**

The improvements the paper needs to make is to compare with LLM-SR, which I think is a very good benchmarks and baseline for scientific discovery.
To the MDP definition, what is your discount factor $\gamma$?

**Reasons To Accept:**

- The idea is clear and noval. It is true that this is the first paper I see to combine DPO with LLM with evolutionary search in scientific discovery.
- The storyline is clear and the method is easier to get understand.
- In the experiment section seems all settings are been listed, which is good for reproduction.

**Reasons To Reject:**

- Lack of a crucial reference. The LLM-SR proposes similar ideas but using training-free methods. Lack of comparison to LLM-SR makes this paper less convincing.
- Lack of baselines. Why only FunSearch? What about other existing methods? What is the performance without RL fine-tuning? Based on Algorithm 1, I believe the RL-Update steps won't be a necessary step if the EvoTune is combined with models like GPT-4o. Without that update, what are the results?
- Compute‑efficiency trade‑off unquantified. The additional RL phase surely adds GPU hours, how to compare that with FUNSEARCH?

[1] Shojaee, P., Meidani, K., Gupta, S., Farimani, A. B., & Reddy, C. K. (2024). Llm-sr: Scientific equation discovery via programming with large language models. arXiv preprint arXiv:2404.18400.

---

> ### Author Response · Authors · 2025-06-01
>
> We thank the reviewer for their thoughtful and detailed feedback. We appreciate the recognition of our work’s **novelty in integrating RL (via DPO) with evolutionary search**. We are glad that the reviewer found the algorithm and experimental setup well motivated, clearly explained and well specified.
>
> Below, we address the specific points raised:
>
> #### **1. Task scope and significance, comparison to LLM-SR**
>
> > *The paper is evaluated only on three well-known combinatorial optimization tasks without any extension to other meaningful tasks. For example, LLM-SR is more on the realistic scenario than this paper.
> …
> The improvements the paper needs to make is to compare with LLM-SR, which I think is a very good benchmarks and baseline for scientific discovery.*
> >
>
> To strengthen the evaluation, we incorporated results on the LLM-SR benchmark, as suggested, and introduced an additional benchmark inspired by real-world engineering problems at Google. Further details can be found in our response above to all reviewers. We hope this addresses the concerns about the significance and generality of our approach.
>
> #### **2. Baselines and ablation study**
>
> > *Lack of baselines. Why only FunSearch? What about other existing methods? What is the performance without RL fine-tuning? Based on Algorithm 1, I believe the RL-Update steps won't be a necessary step if the EvoTune is combined with models like GPT-4o. Without that update, what are the results?*
> >
>
> We agree on the importance of comprehensive baselines, therefore, we invite you to read our response to all reviewers for problem-specific baselines other than FunSearch.
>
> Our use of FunSearch as the primary baseline was intended to provide a direct comparison to the leading LLM-based evolutionary approach. To clarify, the “without RL-Update” variant requested by the reviewer is ***precisely*** the FunSearch baseline reported in our experiments. This comparison allows us to isolate the impact of our RL-based fine-tuning component relative to evolutionary search alone, and we hope this resolves any doubts about the fairness and completeness of our comparisons.
>
> We agree with the reviewer that evaluating EvoTune on frontier models such as GPT-4o would be highly valuable and of significant interest. However, without access to the weights of these proprietary models, we are cannot assess whether RL training would yield improvements in steering the discovery toward better regions of the search space. Nonetheless, we hope that our results across three different open-source models in the main paper already demonstrate the robustness and generality of our method beyond a single model.
>
> Additionally, as with most LLM tasks, scaling model size is helpful, and we believe EvoTune and FunSearch are no exceptions to this trend. While our experiments compare models of the same scale, we conjecture the benefits of integrating RL with evolutionary search would persist even when scaling to frontier models.
>
> ####  **3. Computational overhead of RL phase**
>
> > *Compute‑efficiency trade‑off unquantified. The additional RL phase surely adds GPU hours, how to compare that with FUNSEARCH?*
> >
>
> To address the reviewer’s concern, we have included runtime comparisons along with a more detailed discussion in our response to all reviewers.
>
> ####  **4. Discount factor ($\gamma$)**
>
> > *To the MDP definition, what is your discount factor $\gamma$?*
> >
>
> In our setup, the discount factor $\gamma$ is set to 1. This is standard in episodic RLHF settings like ours, where rewards are provided only at the end of each episode (i.e., upon solution evaluation). We will clarify this explicitly in the manuscript.

---

> > ### Comment · Reviewer_rxQm · 2025-06-04
> >
> > Thanks for your response.
> > >To strengthen the evaluation, we incorporated...
> >
> > You may also want to include how LLM-SR performs in these tasks. The key insight is not about the LLM-SR task, but also how your method compares to LLM-SR.
> >
> > >  To clarify, the “without RL-Update” variant requested by the reviewer is precisely the FunSearch baseline reported..
> >
> > Sorry for the confusion. What I mean is how your method compares to proprietary models. My question should be *"if the **FunSearch** is combined with models like GPT-4o."*  For example, the improvements of RL fine-tuning over the non-fine-tuning version are indeed since you include the extra training steps. But how does it compare to the GPT4-o FunSearch? Will the RL result beat GPT4-o? These experiments will be helpful. It also might be helpful to compare the fine-tune 1B version to the Funsearch 3B version. My key concern is, RL does improve the performance, but will the extra training be worth it? If 3B version is better than the 1B fine-tuned version, how can to justify the necessity of fine-tuning?
> >
> > I am also confused about the generalization ability of this method. Will the RL-finetuned version have worse generation ability on other datasets? To further clarify, the generation ability is, if you fine-tune your method on traveling salesmen, will it still perform good on BinPack (FunSearch)? Will there be a domain overfitting problem?
> >
> > To summarize, my major concern is, fine-tuning always improves the model performance, which does not seem to be a good contribution.
> >
> > Thanks a lot for your external experiments and looking forward if some of the results will be presented. Sorry again for my confusing question in the original reviews. Also, please let me know if any of my questions or understanding is wrong. I am happy to increase my score if my major concern gets resolved.

---

> > ### Author Response · Authors · 2025-06-06
> > **Response [1/3]**
> >
> > We thank the reviewer for their follow-up, clarifications, and openness to revisiting their evaluation. We appreciate the opportunity to address these points directly, please find our responses below.
> >
> > #### **1. Comparison to LLM-SR**
> >
> > > *You may also want to include how LLM-SR performs in these tasks. The key insight is not about the LLM-SR task, but also how your method compares to LLM-SR.*
> > >
> >
> > The LLM-SR paper indeed introduces both a (i) benchmark for symbolic regression and (ii) a method for solving it. The core methodology of LLM-SR, however, closely parallels FunSearch -  LLM-SR utilizes an LLM-based evolutionary search framework and adapts it for symbolic regression tasks.
> >
> > As acknowledged by the LLM-SR authors:
> >
> > > *Most related to our work is FunSearch (Romera-Paredes et al., 2024) that combines LLMs with systematic evaluators to search for heuristics... Building upon these ideas, our LLM-SR framework employs LLM as an optimizer...*
> > >
> >
> > The LLM-SR method can be understood as a specialization of FunSearch for symbolic regression, with the following task-specific adjustments:
> >
> > - The *prompt* contains a problem specification (concise description of the scientific problem).
> > - The *generated program* is specifically an equation function.
> > - *Evaluation* includes optimization of equation parameters on a problem-specific dataset, using either stochastic gradient descent or BFGS, as described in their work (Torch/Numpy versions).
> >
> > Another key shared element is the program database (which LLM-SR refers to as an "experience buffer"). This component is structurally identical to the one used in the original FunSearch paper and in our work. This commonality encompasses both the architecture (island-based with clusters within islands) and the sampling methodology (uniform random island selection, Boltzmann sampling prioritizing higher-scoring clusters, and a preference for shorter programs when sampling within clusters).
> >
> > Given these methodological similarities, the FunSearch baseline we employed in our experiments already includes the core techniques underlying LLM-SR. Thus, we believe that our existing reported improvements of EvoTune over the FunSearch baseline implicitly demonstrate that adding RL finetuning would benefit an LLM-SR-like approach. In summary, the baseline we call “FunSearch” is essentially the LLM-SR method, and “EvoTune” can be interpreted as combining the LLM-SR method with RL finetuning.
> >
> > However, to fully address the reviewer’s request for comparison with LLM-SR, we ran experiments on two of their benchmark tasks (*E. coli Growth* and *Stress-Strain*), under comparable compute budgets of ~10,000 sampled programs (as reported as 2,500 iterations in the LLM-SR paper). We report the Normalized MSE (lower is better) of the best-performing function found on the train set, evaluated on both in-distribution (ID) and out-of-distribution (OOD) test datasets from LLM-SR.
> >
> > | Method | E. coli Growth (ID) | E. coli Growth (OOD) | Stress-Strain (ID) | Stress-Strain (OOD) |
> > | --- | --- | --- | --- | --- |
> > | EvoTune (Phi 3.5 Mini 3.8B) | 0.0082 | 0.0322 | **0.0033** | **0.0035** |
> > | FunSearch [LLM-based evolutionary search in our implementation] (Phi 3.5 Mini 3.8B) | 0.0383 | 0.0636 | 0.0037 | 0.0074 |
> > | LLM-SR (Mixtral 8x7B) | **0.0026** | **0.0037** | 0.0162 | 0.0946 |
> > | LLM-SR (GPT-3.5-turbo) | 0.02140 | 0.0264 | 0.0210 | 0.0516 |
> >
> > We hope these results demonstrate the strength of our pipeline and address the reviewer’s request for comparison to LLM-SR. **Furthermore, these results show that EvoTune, using the relatively small Phi-3 Mini (3.8B) model, is competitive and, in the case of the Stress-Strain problem, outperforms both the Mixtral 8x7B and GPT-3.5-turbo versions of LLM-SR, even on this specialized benchmark.** This highlights the efficacy of EvoTune's RL-based fine-tuning.
> >
> > We are encouraged that EvoTune demonstrates strong performance even within LLM-SR's specific domain. Additionally, we emphasize that while the LLM-SR method is targeted to symbolic regression, **EvoTune is a more general framework** -  it can be applied to any problem where solutions are readily verifiable with a sufficiently dense feedback signal and can be structured around a code skeleton. It supports a broad class of reward signals, e.g., optimality gap, runtime, memory use, or even verbal feedback of a judge LLM translated to some scalar range. As such, EvoTune is designed for much wider applicability in scientific discovery and optimization.

---

> > ### Author Response · Authors · 2025-06-06
> > **Response [2/3]**
> >
> > #### **2. Comparisons to Larger or Proprietary Models (e.g., GPT-4o, 3B vs 1B)**
> >
> > > *Sorry for the confusion. What I mean is how your method compares to proprietary models. My question should be *"if the **FunSearch** is combined with models like GPT-4o."* For example, the improvements of RL fine-tuning over the non-fine-tuning version are indeed since you include the extra training steps. But how does it compare to the GPT4-o FunSearch? Will the RL result beat GPT4-o? These experiments will be helpful.*
> > >
> >
> > Regarding comparing to a frontier model like GPT-4o, we would like to clarify that **EvoTune is a technique designed to improve the performance of the underlying language model, whatever that model may be.**
> >
> > If we had access to GPT-4o and the resources for extensive experimentation, the most direct test of EvoTune's added value would be to compare **GPT-4o + FunSearch** against **GPT-4o + EvoTune**. If GPT-4o + EvoTune showed improvements, it would demonstrate EvoTune's ability to further boost even the most capable models. While we don't claim that EvoTune on a small open-sourced model surpasses GPT-4o + FunSearch on sample efficiency (as the base model's capability is an important factor), we hypothesize that **GPT-4o + EvoTune could outperform GPT-4o + FunSearch**.
> >
> > Furthermore, as a relevant data point, for the LLM-SR task, our results show that EvoTune with Phi 3.5 Mini 3.8B model can already outperform GPT-3.5-Turbo + FunSearch.
> >
> > We also note that querying proprietary models many times, as is required for evolutionary search methods, can be expensive. For example, in the FunSearch paper, which uses a proprietary model, Codey, they sample on the order of a million functions (a conservative back-of-the-envelope estimate suggests that a single instance of evolutionary search with 1 million program prompts costs ~$5-15k), which is **prohibitively expensive** for many scientific use cases, especially for academic labs.
> >
> > > *It also might be helpful to compare the fine-tune 1B version to the Funsearch 3B version. My key concern is, RL does improve the performance, but will the extra training be worth it? If the 3B version is better than the 1B fine-tuned version, how can to justify the necessity of fine-tuning?*
> > >
> >
> > The reviewer's suggestion to compare 3B+FunSearch to 1B+EvoTune likely comes from understanding that the RL finetuning step presents an extra computational load. The reviewer proposes to offset the extra load by sampling from a larger, more resource-intensive base model. As other reviewers raised a similar concern on the extra computational load of the RL phase, we address this point in the **Computational overhead of RL phase** section in our common response to all reviewers. We invite the reviewer to consult the section where we show that EvoTune can not only improve sample efficiency but also wall-clock efficiency over the FunSearch baseline for the same underlying model. Moreover, we point out that using larger models could actually yield worse wall-clock efficiency. Although larger models often produce higher-quality outputs on average, they also require more time to generate each sample. As a result, fewer samples can be obtained within a fixed time frame, in addition to requiring more GPU memory.
> >
> > Additionally, when comparing larger models to smaller ones, Table 1 in the original manuscript reveals an noteworthy trend: both FunSearch and EvoTune variants of Granite model (2B parameters) outperform the FunSearch variant of Phi models (3.8B parameters) across nearly all tasks and data splits, despite Granite being roughly half the size of Phi. A similar trend can even be observed when comparing Llama EvoTune (1B) with Phi FunSearch on some problems and data splits. These trends show that the base model’s size is not necessarily the determining factor, and EvoTune consistently boosts the performance of any base model. We hope this helps address the question about the scale of the base model used with FunSearch.
> >
> > In summary, the **core contribution of EvoTune is orthogonal to the underlying model capabilities and scale**, it provides a lightweight, model-agnostic framework to steer generation toward high-performing programs, regardless of model size. While scaling remains a valuable research direction, we demonstrate that RL-based updates bring measurable improvements even in small models, without relying on parameter count or pretraining data to carry the search.

---

> > ### Author Response · Authors · 2025-06-06
> > **Response [3/3]**
> >
> > #### **3. “Fine-tuning Always Helps”**
> >
> > > *my major concern is, fine-tuning always improves the model performance, which does not seem to be a good contribution*
> > >
> >
> > While we recognize that fine-tuning is broadly beneficial, the assertion that it "always improves performance" does not fully account for the nuanced dynamics of evolutionary search. In the context of evolutionary search, it is *not* straightforward that fine-tuning is always beneficial - overfitting to past good solutions can reduce exploration and harm long-term discovery of better functions.
> >
> > This is precisely why we frame the update as an RL objective (specifically DPO) rather than traditional supervised fine-tuning. Additionally, we show that RL training does not merely increase the average score of the population, but extends the “tail” of the distributions towards higher scores.
> >
> > Moreover, as diversity collapse is commonly observed in training LLMs with RLHF methods, another contribution in our work shows that using forward KL divergence to regularize against the reference policy outperforms using reverse KL divergence (which is commonly used in DPO). This is likely due to its mean-seeking behaviour, which avoids mode collapse and preserves the creativity of LLM outputs, which are crucial for evolutionary search to work well.
> >
> > Thus, we believe our contribution lies in how we fine-tune, not just in the fact that we do, and in showing that principled reward-based updates yield consistent performance gains in evolutionary LLM search pipelines.
> >
> > #### **4. Generalization and Overfitting Across Domains**
> >
> > > *I am also confused about the generalization ability of this method. Will the RL-finetuned version have worse generation ability on other datasets? To further clarify, the generation ability is, if you fine-tune your method on traveling salesmen, will it still perform good on BinPack (FunSearch)? Will there be a domain overfitting problem?*
> > >
> >
> > Similarly to FunSearch, LLM-SR, and other methods in automatic algorithm discovery, our approach is **task-specific by design**. We do **not aim for cross-task generalization** — e.g., transferring from TSP to Bin Packing — unless explicitly adapted for it. **Generalization in the context of discovery methods like EvoTune is usually considered across instances within a task (e.g., different sizes of TSP), not across unrelated tasks**. This is essential when tackling diverse domains such as combinatorial optimization, symbolic regression, or biological modeling. Since finetuning with LoRA adapters is relatively cheap, each task is tackled separately.
> >
> > That said, our formulation naturally supports continual fine-tuning across domains, as rewards can be aggregated online. We see this as a promising avenue for future work in lifelong scientific discovery and highlight that our method provides a unified framework to make this tractable, unlike supervised approaches, which require new ground truth for every domain shift.
> >
> > We thank the reviewer again for their thoughtful engagement and detailed follow-up questions. We hope this response has clarified the key points raised, including the nature of our baselines and LLM-RS comparison, the role of model size, and the scope of generalization. If this addresses the concerns of the reviewer, we would sincerely appreciate the consideration in raising the score.

---

> > > ### Comment · Reviewer_rxQm · 2025-06-07
> > >
> > > Thanks for your clarification. I think my questions have been properly addressed. However, the only concern is the statement *"In the context of evolutionary search, it is not straightforward that fine-tuning is always beneficial - overfitting to past good solutions can reduce exploration and harm long-term discovery of better functions."* Do you have anything to support this statement?

---

> > ### Author Response · Authors · 2025-06-07
> >
> > Thank you for your follow-up question. We appreciate the opportunity to provide further support for our statement:  *"In the context of evolutionary search, it is not straightforward that fine-tuning is always beneficial - overfitting to past good solutions can reduce exploration and harm long-term discovery of better functions."*
> >
> > This statement is grounded in one of the fundamental challenges in evolutionary algorithms - **premature convergence** [1]**.** This occurs when an evolutionary algorithm converges to a suboptimal solution, losing the diversity necessary to explore the search space and find the global optimum.
> >
> > An overly aggressive fine-tuning strategy can "overfit" current solutions, sacrificing the exploration needed for long-term discovery. This is a classic problem of over-exploitation at the expense of exploration. Our experiments confirmed this empirically. If the learning rate was set too high, the model stopped exploring and began repeating previous good solutions. The best program scores stagnated, and the discovery of new, unique solutions (measured by the number of unique scores) came to a halt. In these cases, the run with improper training was quickly surpassed by a baseline run with no training at all, demonstrating that naive fine-tuning can be detrimental.
> >
> > This highlights that our contribution is not just in the fact that we finetune, but also in finding the right **recipe** for training. Our choice of a DPO algorithm with a forward KL regularizer preserved the diversity needed to improve top-performing solutions while guiding the search toward higher-scoring regions. For further evidence, we direct the reviewer to Appendix A.11, where we tried an alternative SFT-based training method, but it did not perform as well.
> >
> > Thank you again for your valuable feedback.
> >
> > [1] Affenzeller, M., Winkler, S., Wagner, S., & Beham, A. (2009). *Genetic algorithms and genetic programming: Modern concepts and practical applications*. CRC Press.

---

> > > ### Author Response · Authors · 2025-06-10
> > >
> > > We sincerely thank the reviewer for their engagement and valuable feedback. We will incorporate the revisions guided by their suggestions. As the discussion period draws to a close, we would be grateful if the reviewer could kindly let us know whether our responses have adequately addressed all their concerns. We remain available to provide any final clarifications.

---

### Official Review · Reviewer_e6jZ · 2025-05-15

**Rating:** 8
**Confidence:** 4
**Ethics Flag:** 1

**Summary:**

The paper targets automated discovery of efficient algorithms and advances prior LLM‑based evolutionary search by making the language model itself a learnable search operator. The proposed framework interleaves evolutionary exploration with reinforcement‑learning fine‑tuning (RL‑FT): newly discovered, higher‑quality candidate algorithms serve as rewards to update the LLM, which in turn generates stronger offspring in subsequent generations. Evaluations on three classic combinatorial‑optimization benchmarks—bin packing, traveling salesman, and flat‑pack assembly—show that the RL‑augmented evolutionary loop finds better algorithms faster than a static‑LLM baseline, demonstrating the practical benefit of coupling RL with evolutionary strategies for algorithm design.

**Reasons To Accept:**

This work has solid motivation, good writing, intensive and well-presented numerical experiments, crystal clear details and presentations, and impressive empirical results.

**Reasons To Reject:**

1. The computation overheads induced by RL fine-tuning of LLMs are not discussed in this paper. A good way to show this is to draw figures similar to Fig. 2 but replace the x-axis as search time.

2. It might have been discussed in the FunSearch paper, but it will be more self-contained if the authors could also compare with non-LLM-based search algorithms. Such algorithms can be more efficient with sampling and thus are more search-time friendly, or not because I can imagine the bottleneck of the search efficiency lies at the long evaluation time of searched programs.

---

> ### Author Response · Authors · 2025-06-01
>
> We are grateful for the recognition of our paper’s solid motivation, comprehensive experiments, and impressive empirical results. We appreciate the reviewer’s positive comments regarding the clarity of our presentation.
>
> Below, we address the two concerns raised:
>
> #### **1. Computational overhead of RL phase**
>
> > *The computation overheads induced by RL fine-tuning of LLMs are not discussed*
> >
>
> We included additional plots where the x-axis reports wall-clock time. Please see our response to all reviewers for the link to the pdf with figures and a discussion of this analysis. We hope this addition provides greater transparency regarding the trade-offs of search efficiency and computational overhead.
>
>
>
> #### **2. Non-LLM-based search algorithms**
>
> > *It might have been discussed in the FunSearch paper, but it will be more self-contained if the authors could also compare with non-LLM-based search algorithms*
> >
>
> We appreciate the suggestion to compare with non-LLM-based methods. To address this, we present comparisons to recently developed neural solvers as well as search-based heuristics (for TSP) and classic human-designed heuristics (for Bin Packing and Flatpack) in our response to all reviewers.

---

> > ### Author Response · Authors · 2025-06-10
> >
> > We thank the reviewer for their valuable insights. As the discussion period is coming to an end, we would appreciate it if the reviewer could kindly let us know whether their concerns have been adequately addressed and whether we could provide any clarifications if there are any outstanding concerns.

---

### Author Response · Authors · 2025-06-01
**Response to all reviewers [1/3]**

We sincerely thank the reviewers for their valuable feedback. The suggestions have been helpful in identifying areas for clarification and improvement. We address the common concerns below. We provide a ***pdf with additional figures requested by reviewers on [this](https://www.dropbox.com/scl/fi/e34kfnuy3jdamnqhxrlpm/COLM-2025-Rebuttal.pdf?rlkey=yfjjr75i23w0588ilbwf223e8&st=8ywm4665&dl=0) link.***

### **1. Computational overhead of RL phase**
> ***Reviewer e6jZ:** The computation overheads induced by RL fine-tuning of LLMs are not discussed in this paper. A good way to show this is to draw figures similar to Fig. 2 but replace the x-axis as search time.*
>
> ***Reviewer rxQm:** Compute‑efficiency trade‑off unquantified. The additional RL phase surely adds GPU hours, how to compare that with FUNSEARCH?*
>
> ***Reviewer YtJU:** The RL fine-tuning introduces additional computational overhead. In addition to reporting the optimality gap, please also report runtime …*
>

Our primary focus in this paper is on sample efficiency. However, upon the reviewers’ requests, we have included comparisons in terms of wall-clock time in Figure 1 of the attached pdf. To ensure a fair evaluation, we have extended FunSearch baseline runs beyond those reported in the main paper.

We are pleased to share that, even with the current unoptimized EvoTune implementation, **our method can already demonstrate improvements in wall-clock efficiency over the FunSearch baseline simply by adjusting existing hyperparameters**. As shown in Figure 2, reducing the RL update frequency (hyperparameter $f_{\text{RL}}$) allows EvoTune to discover superior algorithms in less time, not just with fewer sampled algorithms. This highlights that improvements are achievable without altering the core method. Importantly, this speedup is achieved **while fully preserving the gains in sample efficiency**.

We would like to emphasize that our current EvoTune implementation has not yet been optimized for wall-clock time. Several implementation aspects presently place its runtime at a disadvantage compared to the non-training FunSearch baseline:
- Inference leverages the highly optimized TGI library, while training uses the less speed-optimized TRL library.
- Due to a dependency mismatch, we use Flash Attention only for inference but not for training (could offer up to a ~3x training speedup [1]).
- Due to hardware limitations, we employ gradient checkpointing during training to manage memory usage, which increases computation time.

Addressing these engineering aspects offers an immediate and actionable way to make our comparison fairer, ensuring that both training and inference are maximally optimized for wall-clock efficiency.

Beyond engineering improvements, we outline several promising directions that would further reduce computational overhead while preserving (or even improving) sample-efficiency gains:

- Studying what subset of generated data is most effective for training and makes each update more impactful.
- Identifying the best LLM checkpoints to be used at the beginning of each training iteration.
- Exploring alternative RLHF training algorithms.

We reiterate that the primary contribution of our work is demonstrating that **RL finetuning makes LLMs better search operators** within an evolutionary search framework. It enables LLMs to discover higher-scoring solutions using fewer function samples compared to a non-trained baseline.

Our results establish the viability of integrating RL into evolutionary search such that:

- The training effectively guides the evolutionary search toward superior solutions (measured by the single best or top-k performance) **rather than merely increasing the average score of the population.**
- **The diversity of sampled functions is maintained,** a critical and non-trivial challenge in RLHF training, which we achieve through techniques like Forward KL regularization.

We posit that demonstrating the gain in sample efficiency is a valuable proof-of-concept. While we openly acknowledge the current computational overhead, our results show that improvements in wall-clock efficiency are achievable.

---

> ### Author Response · Authors · 2025-06-01
> **Response to all reviewers [2/3]**
>
> ### **2. Non-LLM baselines**
>
> > ***Reviewer e6jZ:** It might have been discussed in the FunSearch paper, but it will be more self-contained if the authors could also compare with non-LLM-based search algorithms. Such algorithms can be more efficient with sampling and thus are more search-time friendly, or not because I can imagine the bottleneck of the search efficiency lies at the long evaluation time of searched programs.*
> >
> > ***Reviewer rxQm:** Lack of baselines. Why only FunSearch? What about other existing methods?*
> >
> > ***Reviewer YtJU:** Missing baselines. For example, comparisons with traditional TSP solvers (e.g., Concorde, LKH-3) and recently proposed neural solvers should be included.*
> >
>
> We thank the reviewers for highlighting the importance of including non-LLM baselines. We provide the requested comparisons along with a perspective on how our results relate to traditional combinatorial optimization methods.
>
>
> #### **TSP comparisons**
>
> We compare to the following non-LLM baselines and search-based algorithms:
>
> - LEHD [2] is a neural solver successor to POMO [3] and Attention Model [4], which augments supervised learning of heuristics with a decoder specialized for TSP
> - KGLS [5] is a search-based baseline - a more advanced version of the Guided Local Search (GLS) introduced in the paper.
> - NeuralGLS [6] is a hybrid model integrating GLS with graph convolutional networks specialized for TSP
>
> We evaluate EvoTune on 29 TSP instances from TSPLib, comprising real-world TSP instances of varying difficulty. This way we could directly compare with the results reported in [2,5,6]. Additionally, this evaluation serves as an additional benchmark for EvoTune to test the robustness and generalization of the evolved heuristics, since no instances from TSPLib are shown to EvoTune during training. We picked the top-performing program evolved by EvoTune and Granite as the base model.
>
> As a reminder, for TSP, EvoTune evolves the heuristic that is used by guided local search (GLS), and GLS operation can be budgeted to call local search $t_{max}$ times. The GLS-based baselines use $t_{max}=1000$. An interesting and important aspect of EvoTune for TSP is that it was designed with efficiency of training and evaluation in mind; therefore, during training, we set $t_{max}=20$ to reduce evaluation time and avoid a bottleneck due to computational constraints. In the table below, we present the performance of EvoTune’s heuristic when deployed with $t_{max}\in\{100,200,1000\}$, i.e., it was never optimized to find heuristics under such budgets.
>
> From the table, it is clear that the evolved heuristic successfully achieves near-optimal performance when it is provided the same (or even less) local search budget as the baselines. It already performs comparably on average to all of the baselines at $t_{max}=100,200$, and outperforms all of the baselines at an equal budget of $t_{max}=1000$.
>
> The average is across the 29 problem instances, but for brevity, we only show 8 representative instances. We will add the full table to the revised manuscript.
>
> | Instance | LEHD | NeuralGLS | KGLS | EvoTune_100 | EvoTune_200 | EvoTune_1000 |  |
> | --- | --- | --- | --- | --- | --- | --- | --- |
> | eil76 | 2.54 | 0.00 | 1.18 | 1.64 | 1.18 | 1.18 |  |
> | kroA100 | 0.12 | 0.03 | 0.06 | 0.02 | 0.02 | 0.02 |  |
> | kroC100 | 0.32 | 1.77 | 0.01 | 1.25 | 0.83 | 0.01 |  |
> | eil101 | 2.31 | 0.36 | 2.07 | 2.07 | 2.07 | 1.78 |  |
> | pr124 | 1.11 | 0.08 | 0.08 | 0.08 | 0.08 | 0.00 |  |
> | pr144 | 0.19 | 0.74 | 0.00 | 0.38 | 0.09 | 0.00 |  |
> | u159 | 1.13 | 0.90 | 0.96 | 1.44 | 1.44 | 0.00 |  |
> | rat195 | 1.42 | 0.48 | 0.97 | 0.91 | 0.91 | 0.66 |  |
> | kroA200 | 0.64 | 0.86 | 0.71 | 0.56 | 0.23 | 0.20 |  |
> | Average | 1.92 | 0.96 | 0.36 | 0.82 | 0.69 | 0.32 |  |
>
>
> #### **Bin Packing and Flatpack comparisons**
>
> For Bin Packing and Flatpack problems, we include direct comparison to human-designed heuristics. For **Bin Packing**, we report results for the well-established **best-fit heuristic**. For **Flatpack**, we use a **greedy heuristic** that promotes compact, efficient placements by placing larger blocks first and maximizing adjacency.
>
> The table below summarizes the optimality gaps for each method. The sampling budget is 22.4k programs for both EvoTune and FunSearch.
>
> |  | Human-designed heuristic | EvoTune | FunSearch |
> | --- | --- | --- | --- |
> | Bin Packing | 5.37 | 2.06 | 2.96 |
> | Flatpack | 0.1092 | 0.0829 | 0.0898  |
>
> Despite relying on smaller open-source LLMs, both EvoTune and FunSearch are able to **outperform human-designed heuristics**, demonstrating the ability of LLM-driven evolutionary search to discover effective algorithmic strategies within a limited search budget.

---

> > ### Author Response · Authors · 2025-06-01
> > **Response to all reviewers [3/3]**
> >
> > #### **Objective and scope of our evaluation**
> >
> > We would like to clarify that the primary objective in this work is not to achieve state-of-the-art task-specific performance on combinatorial optimization benchmarks, but rather to use these tasks as **testbeds** for rigorously evaluating the effectiveness of our proposed EvoTune method. By focusing on controlled comparisons against FunSearch, we aim to isolate and demonstrate the practical impact of interleaving RL fine-tuning with evolutionary search in the context of algorithm discovery.
> >
> > #### **Broader perspective of the comparison to traditional solvers**
> >
> > While LLM-based evolutionary search is not intended to outright replace specialized classical solvers, it offers several advantages that we believe are complementary and valuable:
> >
> > - **Flexibility and generality**: EvoTune and FunSearch can be applied to a very wide range of optimization problems, including those with complex or non-standard constraints, where it’s common to rely on hand-crafted heuristics or where solvers are not available at all.
> > - **Scalability**: EvoTune and FunSearch operate by searching over algorithmic function space, rather than solving large-scale optimization problems directly. As noted in the FunSearch paper, this leads to greater scalability on large instances where classical solvers struggle.
> > - **Interpretability**: The discovered solutions are **algorithms** rather than black-box numerical outputs. This makes it easier to identify patterns or structure in how solutions are formed - an insight that the FunSearch paper leveraged to restrict the search space and uncover even better solutions.
> >
> > ### **3. Evaluating the method on additional tasks**
> >
> > > ***Reviewer rxQm:** The paper is evaluated only on three well-known combinatorial optimization tasks without any extension to other meaningful tasks. For example, LLM-SR is more on the realistic scenario than this paper.*
> > >
> > > ***Reviewer YtJU:** The paper evaluates the proposed approach on only three CO tasks. Including a broader set of CO or non-CO tasks would strengthen the evaluation.*
> > >
> >
> > We appreciate the reviewers’ suggestion to evaluate our method on a broader set of tasks. While our current study focuses on three combinatorial optimization problems $-$ bin packing, traveling salesman, and flat pack $-$ we selected these tasks for their recognized significance and direct relevance to numerous real-world applications. For example, the Traveling Salesman Problem (TSP) is fundamental to areas such as logistics, network design, and resource allocation, while bin packing is applicable in operations research and industry settings.
> >
> > To further address these concerns and strengthen our evaluation, we evaluated our method on:
> >
> > - **HashCode programming competition questions** [7] inspired by real-world engineering challenges faced at Google (Figure 3 Top)
> > - **LLM-SR** [8] (Figure 3 Bottom)
> >
> > Our expanded experiments are consistent with the results in the original submission, but they also expand the scope and provide additional evidence on the applicability and robustness of our method. We hope this addresses the reviewers’ concerns regarding the diversity and significance of the evaluation tasks.
> >
> >
> > \
> > [1] Dao, T. (2024). FlashAttention-2: Faster Attention with Better Parallelism and Work Partitioning. International Conference on Learning Representations.
> >
> > [2] Luo, F., Lin, X., Liu, F., Zhang, Q., & Wang, Z. (2023). Neural combinatorial optimization with heavy decoder: Toward large scale generalization. Advances in Neural Information Processing Systems.
> >
> > [3] Kwon, Y. D., Choo, J., Kim, B., Yoon, I., Gwon, Y., & Min, S. (2020). Pomo: Policy optimization with multiple optima for reinforcement learning. Advances in Neural Information Processing Systems.
> >
> > [4] Kool, W., Van Hoof, H., & Welling, M. (2018). Attention, learn to solve routing problems!. arXiv preprint arXiv:1803.08475.
> >
> > [5] Arnold, F., & Sörensen, K. (2019). Knowledge-guided local search for the vehicle routing problem. Computers & Operations Research
> >
> > [6] Sui, J., Ding, S., Xia, B., Liu, R., Bu, D. (2024). NeuralGLS: Learning to guide local search with graph convolutional network for the traveling salesman problem. Neural Computing and Applications
> >
> > [7] Veličković, P., Vitvitskyi, A., Markeeva, L., Ibarz, B., Buesing, L., Balog, M., & Novikov, A. (2024). Amplifying human performance in combinatorial competitive programming. arXiv preprint arXiv:2411.19744.
> >
> > [8] Shojaee, P., Meidani, K., Gupta, S., Farimani, A. B., & Reddy, C. K. (2024). Llm-sr: Scientific equation discovery via programming with large language models. arXiv preprint arXiv:2404.18400.

---

### Decision · Program_Chairs · 2025-07-08

**Decision:**

Accept

**Comment:**

This paper proposes a new method that combines evolutionary search and reinforcement learning to help large language models discover algorithms. The initial reviews were positive about the paper's clarity and novel idea. However, all reviewers raised important questions. They were concerned about the computational overhead from the reinforcement learning phase, the limited comparisons to other methods, and the scope of the evaluation.

I was particularly impressed with the authors' rebuttal, which was not just a clarification but included substantial new experimental results. (and as I've checked COLM policy, it seems adding new experimental results is allowed.) They provided the requested wall-clock time analysis, benchmarked against strong non-LLM baselines, and demonstrated the method's effectiveness on entirely new task domains. This thorough response resolved the reviewers' initial concerns and significantly strengthened the paper. The AC decide to recommend Accept.